# Evidential Softmax for Sparse Multimodal Distributions in Deep Generative Models

**Phil Chen, Masha Itkina, Ransalu Senanayake, Mykel J. Kochenderfer**
Stanford University
{philhc, mitkina, ransalu, mykel}@stanford.edu

## Abstract

Many applications of generative models rely on the marginalization of their high-dimensional output probability distributions. Normalization functions that yield sparse probability distributions can make exact marginalization more computationally tractable. However, sparse normalization functions usually require alternative loss functions for training since the log-likelihood is undefined for sparse probability distributions. Furthermore, many sparse normalization functions often collapse the multimodality of distributions. In this work, we present *ev-softmax*, a sparse normalization function that preserves the multimodality of probability distributions. We derive its properties, including its gradient in closed-form, and introduce a continuous family of approximations to *ev-softmax* that have full support and can be trained with probabilistic loss functions such as negative log-likelihood and Kullback-Leibler divergence. We evaluate our method on a variety of generative models, including variational autoencoders and auto-regressive architectures. Our method outperforms existing dense and sparse normalization techniques in distributional accuracy. We demonstrate that *ev-softmax* successfully reduces the dimensionality of probability distributions while maintaining multimodality.

## 1 Introduction

Learning deep generative models over discrete probability spaces has enabled state-of-the-art performance across tasks in computer vision [1], natural language processing [2]–[4], and robotics [5]–[7]. The probability distributions of deep generative models are often obtained from a neural network using the *softmax* transformation. Since the *softmax* function has full support, the logarithm of the output probabilities is well-defined, which is important for common probabilistic loss functions such as the negative log likelihood, cross entropy, and Kullback-Leibler (KL) divergence.

Several applications highlight the importance of achieving sparsity in high dimensional dense distributions to make downstream tasks computationally feasible and perhaps even interpretable [8]–[11]. In discrete variational autoencoders (VAEs), for instance, calculating the posterior probabilities exactly requires marginalization over all possible latent variable assignments, which is intractable for large latent spaces [9], [10]. Stochastic approaches to circumvent exact marginalization include the score function estimator [12], [13], which suffers from high variance, and continuous relaxations of the discrete latent variable, such as Gumbel-Softmax [14], [15], which introduce bias.

Alternatively, high-dimensional latent spaces can be tractably marginalized using normalization functions that produce sparse probability distributions. Several sparse alternatives to *softmax* have been proposed, including *sparsemax* [8], $\alpha$-*entmax* [16], and *sparsehourglass* [17]. However, such approaches often collapse the multimodality of distributions, resulting in unimodal probability distributions [10] (see Fig. 1). When marginalizing over discrete latent spaces, maintaining this multimodality is crucial for applications such as image generation, where the latent space is multimodal in nature [1], and machine translation, where multiple words can be valid translations and require

35th Conference on Neural Information Processing Systems (NeurIPS 2021).

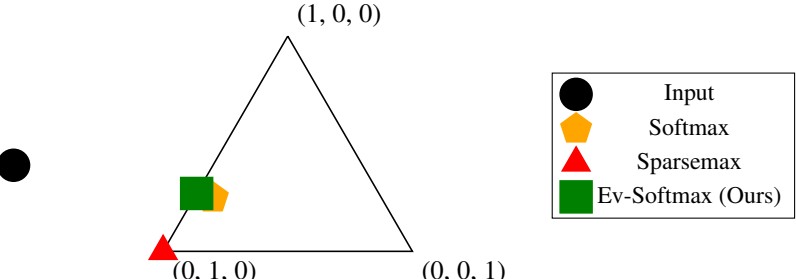

Figure 1: We evaluate the *softmax*, *sparsemax*, and *ev-softmax* (ours) functions on an example input point $\mathbf{v} \in \mathbb{R}^3$ and plot the resulting values on the plane $v_1 + v_2 + v_3 = 1$. We plot the points $\mathbf{v} = (0.4, 1.4, -0.8)$, *softmax*$(\mathbf{v}) \approx (0.25, 0.67, 0.07)$, *sparsemax*$(\mathbf{v}) = (0, 1, 0)$, and *ev-softmax*$(\mathbf{v}) \approx (0.27, 0.73, 0)$. *Sparsemax* yields a unimodal distribution, while *ev-softmax* yields a bimodal distribution which is close to that of *softmax*.

context to distinguish between the optimal word choice [9]. Furthermore, these approaches either require the construction of alternative loss functions to the negative log-likelihood, which is undefined for zero-valued output probabilities [8], [16]–[18], or they are applied as a post-hoc transformation at test time [10]. For the latter case, the latent distributions obtained through these existing sparse normalization functions and their resulting posteriors are not trained directly.

The post-hoc sparsification procedure introduced by Itkina *et al.* [10] provides a method for obtaining sparse yet multimodal distributions at test time. We generalize this method to a sparse normalization function termed *evidential softmax* (*ev-softmax*) which can be applied during both training and test time. We define continuous approximations of this function that have full support and are thus compatible with typical loss terms, such as negative log-likelihood and KL divergence. We use this full-support approximation of *ev-softmax* to directly optimize discrete generative models for objective functions with probabilistic interpretations. This approach is in contrast with post-hoc sparsification methods that optimize a proxy model [10] and methods that use non-probabilistic objective functions, such as *sparsemax* [8]. We evaluate these discrete generative models at test time with the *ev-softmax* function to obtain sparse yet multimodal latent distributions.

**Contributions:** This paper proposes a strategy for training neural networks with sparse probability distributions that is compatible with negative log-likelihood and KL-divergence computations. We generalize the evidential sparsification procedure developed by Itkina *et al.* [10] to the *ev-softmax* function. We prove properties of *ev-softmax* and its continuous approximation, which has full support. Our approach outperforms existing dense and sparse normalization functions in distributional accuracy across image generation and machine translation tasks.

## 2 Background

**Notation:** We denote scalars, vectors, matrices, and sets as $a$, $\mathbf{a}$, $\mathbf{A}$, and $\mathcal{A}$, respectively. The vectorized Kronecker delta $\boldsymbol{\delta}_i$ is defined to be 1 at the $i$th index and 0 at all the other indices. The $K$-dimensional simplex is written as $\Delta^K := \{\boldsymbol{\xi} \in \mathbb{R}^{K+1} : \mathbf{1}^T\boldsymbol{\xi} = 1, \xi_i \geq 0\}$. The KL divergence of probability distribution $\mathbf{p}$ from distribution $\mathbf{q}$ is denoted as $\mathrm{KL}(\mathbf{p} \mid\mid \mathbf{q})$.

### 2.1 Discrete Variational Autoencoders

To model a data generating distribution, we consider data $\mathbf{x} \in \mathbb{R}^d$, categorical input variable $\mathbf{y} \in \mathcal{Y}$, categorical latent variable $\mathbf{z} \in \mathcal{Z}$, and parameters $\theta \in \mathbb{R}^p$. In the VAE model, $\mathbf{z}$ is sampled from a prior $p_\theta(\mathbf{z})$, and we generate $\mathbf{x}$ conditioned on $\mathbf{z}$ as $p_\theta(\mathbf{x} \mid \mathbf{z})$. The conditional variational autoencoder (CVAE) [19] extends this framework by allowing for the generative process to be controlled by input $\mathbf{y}$. The latent variable $\mathbf{z}$ is thus conditioned on the input $\mathbf{y}$ in the prior $p_\theta(\mathbf{z} \mid \mathbf{y})$.

The VAE estimates parameters $\phi$ and an auxiliary distribution $q_\phi(\mathbf{z} \mid \mathbf{x})$ to calculate the evidence lower bound (ELBO) of the log-likelihood for the observed data [20]:

$$\log p_\theta(\mathbf{x}) \geq \mathbb{E}_{q_\phi(\mathbf{z}|\mathbf{x})} \log p_\theta(\mathbf{x} \mid \mathbf{z}) - \mathrm{KL}(q_\phi(\mathbf{z} \mid \mathbf{x}) \mid\mid p_\theta(\mathbf{z})) \tag{1}$$

The CVAE estimates parameters $\phi$ of an auxiliary distribution $q_\phi(\mathbf{z} \mid \mathbf{x}, \mathbf{y})$ to obtain the ELBO [19]:

$$\log p_\theta(\mathbf{x} \mid \mathbf{y}) \geq \mathbb{E}_{q_\phi(\mathbf{z}|\mathbf{x},\mathbf{y})} \log p_\theta(\mathbf{x} \mid \mathbf{z}) - \mathrm{KL}(q_\phi(\mathbf{z} \mid \mathbf{x}, \mathbf{y}) \mid\mid p_\theta(\mathbf{z} \mid \mathbf{y})). \tag{2}$$

The subsequent discussion focuses on VAEs, and analogous expressions for CVAEs may be obtained by replacing $p_\theta(\mathbf{x})$, $q_\phi(\mathbf{z} \mid \mathbf{x})$, and $p_\theta(\mathbf{z})$ with $p_\theta(\mathbf{x} \mid \mathbf{y})$, $q_\phi(\mathbf{z} \mid \mathbf{x}, \mathbf{y})$, and $p_\theta(\mathbf{z} \mid \mathbf{y})$, respectively.

Maximizing the VAE ELBO using gradient-based methods involves computing the term $\nabla_\phi \mathbb{E}_{q_\phi(\mathbf{z}|\mathbf{x})} \log p_\theta(\mathbf{x} \mid \mathbf{z})$ in Eq. (1), which requires marginalization over the auxiliary distribution $q_\phi(\mathbf{z} \mid \mathbf{x})$. Many Monte Carlo approaches estimate this quantity without explicit marginalization over all possible latent classes $\mathbf{z} \in \mathcal{Z}$. The score function estimator (SFE) is an unbiased estimator which uses the identity $\nabla_\phi \mathbb{E}_{q_\phi(\mathbf{z}|\mathbf{x})} \log p_\theta(\mathbf{x} \mid \mathbf{z}) = \mathbb{E}_{q_\phi(\mathbf{z}|\mathbf{x})} p_\theta(\mathbf{x} \mid \mathbf{z}) \nabla_\phi \log q_\phi(\mathbf{z} \mid \mathbf{x})$ to estimate the gradient through samples drawn from $q_\phi(\mathbf{z} \mid \mathbf{x})$. However, the SFE is prone to high variance, and in practice, sampling-based approaches generally require variance reduction methods such as baselines, control variates, and Rao-Blackwellization techniques [21]. Alternatively, a low-variance gradient estimate can be obtained by the *reparameterization trick*, which separates the parameters of the distribution from the source of stochasticity [20]. The Gumbel-Softmax and Straight-Through reparameterizations rely on a continuous relaxation of the discrete latent space [14], [15]. In contrast to these stochastic approaches, our method, like *sparsemax* [8], $\alpha$-*entmax* [16], and sparsehourglass [17], focuses on obtaining sparse auxiliary distributions $q_\phi(\mathbf{z} \mid \mathbf{x})$ to enable exact marginalization over all latent classes $\mathbf{z} \in \mathcal{Z}$ for which the auxiliary distribution has nonzero probability mass.

## 2.2 Normalization Functions

A common normalization function that maps vectors $\mathbf{v} \in \mathbb{R}^K$ to probability distributions in $\Delta^{K-1}$ is the *softmax* function:

$$\text{SOFTMAX}(\mathbf{v})_k \propto e^{v_k}. \tag{3}$$

One limitation of this function is the resulting probability distribution always has full support, *i.e.* $\text{SOFTMAX}(\mathbf{v})_k \neq 0$ for all $\mathbf{v}$ and $k$. Thus, if the distributions $q_\phi$ in Eq. (1) and Eq. (2) are outputs of the *softmax* function, then calculating the expectations over $q_\phi$ exactly requires marginalizing over all one-hot vectors $\mathbf{z} \in \mathbb{R}^K$, which can be computationally intractable when $K$ is large [9], [10].

Recent interest in sparse alternatives has led to the development of many new sparse normalization functions [8], [10], [16]–[18]. The *sparsemax* function projects inputs onto the simplex to achieve a sparse distribution [8]. The $\alpha$-*entmax* family of functions generalizes the *softmax* and *sparsemax* functions through a learnable parameter $\alpha$ which tunes the level of sparsity [16]. *Sparsehourglass* is another generalization of *sparsemax* with a controllable level of sparsity [17]. Despite the controls, we find that these methods achieve sparsity at the expense of collapsing valid modes.

## 2.3 Post-Hoc Evidential Sparsification

Of particular interest is a sparse normalization function that maintains multimodality in probability distributions. Balancing sparsity and multimodality has been shown to be beneficial for tasks such as image generation [10] or machine translation [16]. Itkina *et al.* [10] propose a post-hoc sparsification method for discrete latent spaces in CVAEs which preserves multimodality in distributions. The method calculates evidential weights as affine transformations of an input feature vector and interprets these weights as evidence either for a singleton latent class $\{z_k\}$ or its complement $\overline{\{z_k\}}$ depending on the sign of the weights. This interpretation, with origins in evidential theory [22], [23], allows for the distinction between conflicting evidence and lack of evidence, which is normally not possible with ordinary multinomial logistic regression. Conflicting evidence can be interpreted as multimodality whereas lack of evidence can correspond to sparsity [10].

For a given dataset and model, the resulting post-hoc sparsification is as follows. A CVAE is trained on a dataset using the *softmax* normalization and the ELBO in Eq. (2). At test time, for a given query $\mathbf{y}$ from the dataset, the learned weights of the last layer of the encoder $\hat{\boldsymbol{\beta}} \in \mathbb{R}^{(J+1) \times K}$, and the corresponding output of the last hidden layer $\boldsymbol{\phi}(\mathbf{y}) \in \mathbb{R}^J$, the evidential weights are:

$$w_k = \hat{\beta}_{0k} - \frac{1}{K} \sum_{\ell=1}^{K} \hat{\beta}_{0\ell} + \sum_{j=1}^{J} \left[ \phi_j \left( \hat{\beta}_{jk} - \frac{1}{K} \sum_{\ell=1}^{K} \hat{\beta}_{j\ell} \right) \right] \tag{4}$$

for each $k \in \{1, \ldots, K\}$. Evidence in support of a latent class $z_k$ then corresponds to $w_k^+ = \max(0, w_k)$ and evidence against it as $w_k^- = \max(0, -w_k)$. The resulting sparse distribution is:

$$p(z_k \mid \mathbf{y}) \propto \mathbb{1}\{m_k \neq 0\} e^{w_k}, \tag{5}$$

where the evidential belief mass $m_k$ is:

$$m_k = e^{-w_k^-} \left( e^{w_k^+} - 1 + \prod_{\ell \neq k} \left( 1 - e^{-w_\ell^-} \right) \right) \tag{6}$$

for each $k \in \{1, \ldots, K\}$. Further information can be found in Appendix A.

## 3 The Evidential Softmax Transformation

Although the evidential sparsification procedure is successful at providing sparse yet multimodal normalizations of scores, it can only be performed post-hoc, without the possibility of being used at training time. Furthermore, it requires $O(JK^2)$ computations to calculate all the weights for each input, with $J$ hidden features in the last layer and $K$ latent classes.

We propose a transformation, which we call *evidential softmax* (*ev-softmax*), on inputs $\mathbf{v} \in \mathbb{R}^K$. We claim that this transformation is equivalent to that of the *post-hoc evidential sparsification* procedure with the advantages of having a simple closed form and requiring only $O(K)$ computations.

Let $\overline{v} = \frac{1}{K} \sum_{i=1}^{K} v_i$ be the arithmetic mean of the elements of $\mathbf{v}$. We define our transformation as,

$$\text{EvSoftmax}(\mathbf{v})_k \propto \mathbb{1}\{v_k \geq \overline{v}\} e^{v_k}. \tag{7}$$

Note that the *post-hoc sparsification* procedure in Section 2.3 takes input vectors $\phi \in \mathbb{R}^J$ and the weights $\hat{\beta} \in \mathbb{R}^{(J+1) \times K}$ of the neural network's last layer whereas the *ev-softmax* function takes as input $\mathbf{v} = \hat{\beta}^T \phi \in \mathbb{R}^K$. We formalize the equivalence of *ev-softmax* with the *post-hoc sparsification* procedure as follows:

**Theorem 3.1.** *For any input vector $\phi \in \mathbb{R}^J$, weight matrix $\hat{\beta} \in \mathbb{R}^{J \times K}$, define the score vector $\mathbf{v} = \hat{\beta}^T \phi$ and variables $w_k$, $w_k^{\pm}$, $m_k$, and $p(z_k)$ as in Section 2.3 for all $k \in \{1, \ldots, K\}$. If the probability distribution of $\mathbf{v}$ is non-atomic, then $\text{EvSoftmax}(\mathbf{v})_k = p(z_k)$ for each $k$.*

The proof of this theorem uses the following two lemmas:

**Lemma 3.2.** *Let $\overline{v} = \frac{1}{K} \sum_{i=1}^{K} v_i$. Then $v_k - \overline{v} = w_k$ for all $k \in \{1, \ldots, K\}$.*

*Proof.* The result follows from algebraic manipulation of Eq. (4). Full details are in Appendix B. □

**Lemma 3.3.** *For $m_k$ as defined in Eq. (6), $\mathbb{1}\{m_k \neq 0\} = \mathbb{1}\{v_k > \overline{v}\}$ for all $k \in \{1, \ldots, K\}$.*

*Proof.* Observe that the condition $\mathbb{1}\{m_k = 0\} = 1 - \mathbb{1}\{m_k \neq 0\}$ is equivalent to the conjunction of the following two conditions: 1) $w_k \leq 0$ and 2) there exists some $\ell \neq k$ such that $w_\ell \geq 0$. From Lemma 3.2, these statements are equivalent to 1) $v_k \leq \overline{v}$ and 2) there exists some $\ell \neq k$ such that $v_\ell \geq \overline{v}$. Note that if $v_k \leq \overline{v}$, then either $v_\ell = \overline{v}$ or there must exist some $\ell \neq k$ such that $v_\ell > \overline{v}$. In both cases, there exists $\ell \neq k$ such that $v_\ell \geq \overline{v}$. Thus, we see that $v_k \leq \overline{v}$ implies there exists some $\ell \neq k$ such that $v_\ell \geq \overline{v}$. We conclude that $\mathbb{1}\{m_k = 0\} = \mathbb{1}\{v_k \leq \overline{v}\}$ and the result follows. □

We briefly sketch the proof of Theorem 3.1 and refer the reader to Appendix B for the full derivation. Lemma 3.2 implies that $e^{w_k} \propto e^{v_k}$. In addition, combining Lemma 3.3 with the non-atomicity of the distribution of $\mathbf{v}$ yields that $\mathbb{1}\{m_k \neq 0\} = \mathbb{1}\{v_k \geq \overline{v}\}$ (Lemma B.3). Eq. (7) follows from these two results.

### 3.1 Properties

Table 1 gives a list of *softmax* alternatives, desirable properties of a normalization function, and which properties are satisfied by each function as inspired by the analysis of Laha *et al.* [17]. Refer to Appendix C for proofs of these properties. These results show that *ev-softmax* satisfies the same properties as the *softmax* transformation. While idempotence can be useful, these normalizations are generally only applied once, and in this regime the idempotence does not provide any further guarantees. Likewise, scale invariance is less applicable in practice because the scale of the inputs to

Table 1: Summary of properties satisfied by normalization functions. Here, ✓ denotes that the property is satisfied. The ∗ denotes that the property is satisfied conditional on either the input or some parameter that is independent of the input. Other properties satisfied by all considered normalization functions include invariance with respect to permutation of the coordinates and existence of the Jacobian. These properties are proven in Appendix C.

| Property | Softmax | Sparsemax [8] | Sparsehourglass [17] | **Ev-Softmax (Ours)** |
|---|---|---|---|---|
| Idempotence | | ✓ | ✓ | |
| Monotonic | ✓ | ✓ | ✓ | ✓ |
| Translation Invariance | ✓ | ✓ | ✓∗ | ✓ |
| Scale Invariance | | | ✓∗ | |
| Full Domain | ✓ | ✓ | ✓ | ✓ |
| Lipschitz | 1 | 1 | $1 + \frac{1}{Kq}$ | $1^*$ |

normalization functions is often one of the learned features of data, and the scale generally provides a measure of confidence in the predictions.

The Jacobians of *softmax* and *ev-softmax* have analogous forms (see Appendix C):

$$\frac{\partial \text{SOFTMAX}(\mathbf{v})_i}{\partial v_j} = \text{SOFTMAX}(\mathbf{v})_i(\delta_{ij} - \text{SOFTMAX}(\mathbf{v})_j) \tag{8}$$

$$\frac{\partial \text{EVSOFTMAX}(\mathbf{v})_i}{\partial v_j} = \text{EVSOFTMAX}(\mathbf{v})_i(\delta_{ij} - \text{EVSOFTMAX}(\mathbf{v})_j). \tag{9}$$

For the Jacobian of the proposed *ev-softmax* function, there are two cases to consider. First, when either or both of $\text{EVSOFTMAX}(\mathbf{v})_i$ and $\text{EVSOFTMAX}(\mathbf{v})_j$ are exactly 0, we have $\frac{\partial \text{EVSOFTMAX}(\mathbf{v})_i}{\partial v_j} = 0$. Intuitively, this occurs because classes that have a probability of 0 do not contribute to the gradient and also remain 0 under small local changes in the input. Second, when both $\text{EVSOFTMAX}(\mathbf{v})_i$ and $\text{EVSOFTMAX}(\mathbf{v})_j$ are nonzero, the partial derivative is equivalent to that of *softmax* over the classes that have nonzero probabilities. Hence, optimizing models using the *ev-softmax* function should yield similar behaviors to those using the *softmax* function.

One notable connection between *ev-softmax* and *sparsemax* is that the supports of both functions can be expressed in terms of mean conditions of their inputs. For an input $\mathbf{v} \in \mathbb{R}^K$, the $i$th class is given zero probability by *ev-softmax* if $v_i < \frac{1}{K}\sum_{k=1}^{K} v_k$. Analogously, denoting the size of the support of *sparsemax* on $\mathbf{v}$ as $\ell$, the $i$th class is given zero probability by *sparsemax* if $v_i \leq -\frac{1}{\ell} + \frac{1}{\ell}\sum_{k=1}^{\ell} v_{(k)}$, where $v_{(k)}$ is the $k$th largest element in $\mathbf{v}$ [8]. Thus, sparsemax considers the arithmetic mean after sparsification, whereas ev-softmax considers the arithmetic mean before sparsification.

Figure 1 presents a visualization of an input point in $\mathbb{R}^3$ and the results upon applying *softmax*, *sparsemax*, and *ev-softmax* to this point. Informally, the interior of the simplex is the set of trimodal distributions, the edges constitute the set of bimodal distributions, and the vertices are the unimodal distributions. Thus, this input provides an example of when *sparsemax* yields a unimodal distribution whereas *ev-softmax* does not. Note that $\mathbb{R}^3$ is the lowest-dimensional space for which the *ev-softmax* function is nontrivial. In $\mathbb{R}^2$, *ev-softmax* is equivalent to the *argmax* function, which is piece-wise constant and has zero gradients everywhere. However, the focus of *ev-softmax* is on balancing multimodality with sparsity for spaces with higher dimension than $\mathbb{R}^2$.

### 3.2 Training-time Modification

Since *ev-softmax* outputs a sparse probability distribution, the negative log-likelihood and KL divergence cannot be directly computed. To address this limitation, we modify *ev-softmax* during training time as follows:

$$\text{EVSOFTMAX}_{\text{train},\epsilon}(\mathbf{v})_i \propto (\mathbb{1}\{v_i \geq \overline{v}\} + \epsilon)e^{v_k} \tag{10}$$

for some small $\epsilon$. In practice, we set $\epsilon$ to $10^{-6}$.

The KL divergence with the above training modification takes the same form as the $\epsilon$-KL used for overcoming sparsity in language modeling [24], [25]. As $\epsilon$ approaches 0, the gradient of the negative

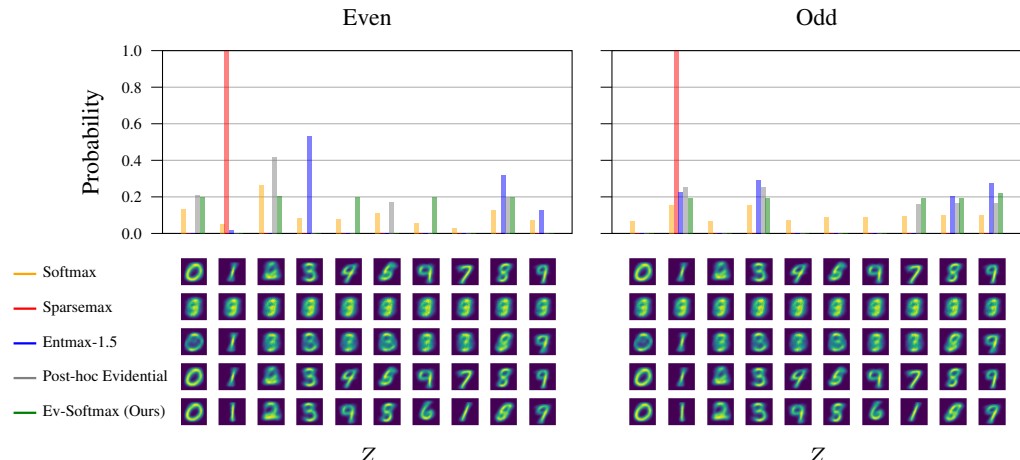

Figure 2: Using the proposed normalization function *ev-softmax* at training time yields a more accurate distribution on the MNIST dataset than *softmax*, *sparsemax*, *entmax*-1.5, and *post-hoc evidential* baselines. The horizontal axis shows the decoded image for each latent class and the vertical axis shows the probability mass. Note that the latent space for the *post-hoc evidential* method is the same as that of *softmax*, since the method is applied only at test time to the latent space distribution. Unlike *ev-softmax*, the sparse distributions produced by *sparsemax* and *entmax*-1.5 collapse the latent space.

log likelihood loss term approaches a form that mirrors that of the *softmax* function:

$$\nabla_{\mathbf{v}} \log\left[\text{SOFTMAX}(\mathbf{v})_i\right] = \boldsymbol{\delta}_i - \text{SOFTMAX}(\mathbf{v}) \tag{11}$$

$$\lim_{\epsilon \to 0} \nabla_{\mathbf{v}} \log\left[\text{EVSOFTMAX}_{\text{train},\epsilon}(\mathbf{v})_i\right] = \boldsymbol{\delta}_i - \text{EVSOFTMAX}(\mathbf{v}), \tag{12}$$

for any finite-dimensional, real-valued vector $\mathbf{v}$. A derivation of Eq. 12 is given in Appendix D.

While the Gumbel-Softmax relaxation also enables gradient computations through discrete spaces, Eq. (12) establishes the convergence and closed-form solution of the gradient as the relaxation goes to 0, which applies uniquely to the *ev-softmax* function. Note that the limit of the log-likelihood of a distribution generated by the training-time modification (i.e. $\lim_{\epsilon \to 0} \log[\text{EVSOFTMAX}_{\text{train},\epsilon}(\mathbf{v})_i]$) is undefined, but Eq. (12) shows that the limit of the *gradient* of the log-likelihood is defined.

## 4   Experiments

We demonstrate the applicability of our proposed strategies on the tasks of image generation and machine translation.[1] We compare our method to the *softmax* function as well as sparse normalization functions *sparsemax* [8] and *entmax*-1.5 [16]. We also compare with the *post-hoc evidential sparsification* procedure [10]. *Sparsehourglass* is not used as it is similar to $\alpha$-*entmax*. Experiments indicate that *ev-softmax* achieves higher distributional accuracy than *softmax*, *sparsemax*, *entmax*-1.5, and *post-hoc evidential sparsification* by better balancing the objectives of sparsity and multimodality.

### 4.1   Conditional Variational Autoencoder (CVAE)

To gain intuition for the *ev-softmax* function, we consider the multimodal task of generating digit images for *even* and *odd* queries ($\mathbf{y} \in \{\text{even}, \text{odd}\}$) on MNIST. We use a CVAE with $|\mathcal{Z}| = 10$ latent classes. Since the true conditional prior distribution given $\mathbf{y} \in \{\text{even}, \text{odd}\}$ is uniform over 5 digits, this task benefits from both sparsity and multimodality of a normalization function.

**Model:** We follow the CVAE architecture and training procedure of Itkina *et al.* [10] for this task. For each of the four normalization functions we consider (*ev-softmax*, *sparsemax*, *entmax*-1.5, and *softmax*), we train an encoder, which consists of two multi-layer perceptrons (MLPs), and a decoder. The first MLP of the encoder receives an input query $\mathbf{y}$ and outputs a categorical prior distribution $p_\theta(\mathbf{z} \mid \mathbf{y})$ over the latent variable $\mathbf{z}$ generated by the normalization function. The second MLP takes as input a feature vector $\mathbf{x}$ in addition to the query $\mathbf{y}$, and outputs the probability distribution for the

---

[1]Code for the experiments is available at `https://github.com/sisl/EvSoftmax`.

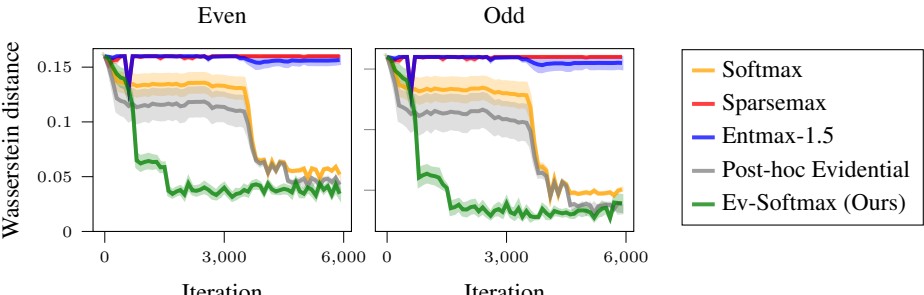

Figure 3: Wasserstein distance between the predicted categorical prior and the ground truth prior distributions. Our method outperforms *softmax*, *sparsemax*, and *entmax*-1.5 over 10 random seeds. Lower is better.

posterior $q_\psi(\mathbf{z} \mid \mathbf{x}, \mathbf{y})$, generated using the same normalization function. The decoder consists of a single MLP that takes as input the latent variable $\mathbf{z}$ and generates an output image $\mathbf{x}$. Each model was trained for 20 epochs to minimize the ELBO of Eq. (2). Further details can be found in Appendix E.

**Results and discussion:** In Fig. 2, we show the decoded images of the latent classes trained using each of the normalization functions, as well as the corresponding prior distributions over these latent classes for even and odd inputs. Using *ev-softmax* at training time yields the most sensible images for both even and odd inputs. The CVAE trained with *ev-softmax* learns a distribution with five nonzero probability modes for both even and odd inputs. All but one of the digits are represented by the latent classes of the *ev-softmax* distribution, and they are generated by the even and odd priors almost perfectly. In contrast, the *softmax* distribution results in nonzero probability mass assigned to all latent classes for both inputs, producing a distributional accuracy that is qualitatively worse than that of *ev-softmax*. While the *post-hoc evidential* distribution also has multiple nonzero probability modes for both even and odd inputs, the distribution is less accurate, demonstrating the advantage of *ev-softmax* in training using the sparse latent distribution. The decoded images of *entmax*-1.5 show that *entmax*-1.5 reduces the posterior latent space to fewer than ten classes, decreasing the representational power of the generative model. Over 10 random seeds and a hyperparameter grid-search, *sparsemax* collapsed both the prior and posterior distributions to a single mode.

To quantify the distributional accuracy of each CVAE model, we compute the Wasserstein distance (Appendix E) between the model's predicted categorical prior $p(\mathbf{z} \mid \mathbf{y})$ and the ground truth prior, shown in Fig. 3. We see that *ev-softmax* converges the fastest and obtains the lowest distance from the ground truth distribution. *Sparsemax* and *entmax*-1.5, on the other hand, do not allow the model to converge toward the true prior as they collapse the prior and posterior distributions. The model trained with *softmax* converges more slowly and to a higher distance from the ground truth distribution because it lacks the sparsity that *ev-softmax* offers, resulting in worse distributional accuracy. The *post-hoc evidential* procedure yields the second lowest Wasserstein distance, but it converges more slowly than *ev-softmax*.

## 4.2 Semi-supervised Variational Autoencoder

To evaluate our method in a semi-supervised setting, we consider the semi-supervised VAE of Kingma *et al.* [26]. We evaluate the classification performance of the model on the MNIST dataset, using 10% of the labeled data and treating the remaining data as unlabeled. In this regime, we expect that sparsity remains a key ingredient since we intuitively associate each digit with one latent class, while multimodality reflects uncertainty in the generative model.

**Model:** We follow the architecture and training procedure of Correia *et al.* [9]. We model the joint probability of an MNIST image $\mathbf{x} \in \mathbb{R}^d$, a continuous latent variable $\mathbf{h} \in \mathbb{R}^n$, and a discrete latent variable $\mathbf{z} \in \{\boldsymbol{\delta_1}, \ldots, \boldsymbol{\delta_{10}}\}$ as $p_\theta(\mathbf{x}) = p_\theta(\mathbf{x} \mid \mathbf{z}, \mathbf{h})p(\mathbf{z})p(\mathbf{h})$. We assume $\mathbf{h}$ has an $n$-dimensional standard Gaussian prior and $\mathbf{z}$ a uniform prior.

We compare *ev-softmax* to other sparse normalization functions, including *sparsemax* and *entmax*-1.5. We also train a model using *softmax* and apply *post-hoc evidential* sparsification at test time. Since we follow the procedure of Correia *et al.* [9], we additionally report the results they obtained for stochastic methods. Unbiased gradient estimators that they report include SFE with a moving average baseline; SFE with a self-critic baseline (SFE+) [27]; neural variational inference and learning (NVIL) with a

learned baseline [28]; and sum-and-sample, a Rao-Blackwellized version of SFE [21]. They also evaluate biased gradient estimators including Gumbel-Softmax and straight-through Gumbel-Softmax (ST Gumbel) [14], [15]. Each model was trained for 200 epochs.

**Results and discussion:** Table 2 shows that using *ev-softmax* yields the highest classification accuracy of models using either Monte Carlo methods or marginalization. Compared to *sparsemax* and *entmax*-1.5, *ev-softmax* maintains higher multimodality in the output distribution, reducing the latent space by 84% on average. In particular, both *sparsemax* and *entmax*-1.5 reduce the latent space by almost 90%, but at the cost of decreased accuracy. On the other hand, *post-hoc evidential sparsification* preserves more latent classes than *ev-softmax* but with a higher variance, and it also achieves a lower accuracy. This suggests that in this semi-supervised setting, training with sparsity and multimodality in the output distribution are both important for distributional accuracy, which benefits overall performance.

Table 2: The results of the semi-supervised VAE on MNIST show that *ev-softmax* yields the best semi-supervised performance. We report the mean and standard error of the accuracy and number of decoder calls for 10 random seeds.

| Method | Accuracy (%) | Decoder calls |
|---|---|---|
| *Monte Carlo* | | |
| SFE | $94.75 \pm .002$ | 1 |
| SFE+ | $96.53 \pm .001$ | 2 |
| NVIL | $96.01 \pm .002$ | 1 |
| Sum&Sample | $96.73 \pm .001$ | 2 |
| Gumbel | $95.46 \pm .001$ | 1 |
| ST Gumbel | $86.35 \pm .006$ | 1 |
| | | |
| *Marginalization* | | |
| Softmax | $96.93 \pm .001$ | 10 |
| Sparsemax | $96.87 \pm .001$ | $1.01 \pm 0.01$ |
| Entmax-1.5 | $97.20 \pm .001$ | $1.01 \pm 0.01$ |
| Post-hoc evidential | $96.90 \pm .001$ | $1.72 \pm 0.05$ |
| **Ev-softmax** | $\mathbf{97.30 \pm .001}$ | $1.64 \pm 0.01$ |

## 4.3 Vector-Quantized Variational Autoencoder (VQ-VAE)

We demonstrate the performance of our approach on a much larger latent space by training a Vector Quantized-Variational Autoencoder (VQ-VAE) model on *tiny*ImageNet [29] for image generation. The VQ-VAE learns discrete latent variables embedded in continuous vector spaces [1]. We use the *tiny*ImageNet dataset due to computational constraints.

**Model:** We train the VQ-VAE architecture in two stages. First the VQ-VAE model is trained assuming a uniform prior over the discrete latent space. For each normalization method, we train an autoregressive prior over the latent space from

Table 3: We sample 25 images from each of the 200 latent classes and compute downstream classification performance (top-5 accuracy and top-10 accuracy) and average number of nonzero latent classes. Our method yields images with the highest top-5 and top-10 accuracy scores. $K$ indicates the average number of nonzero latent classes.

| Method | Acc-5 | Acc-10 | $K$ |
|---|---|---|---|
| Softmax | $38.4 \pm 0.2$ | $48.8 \pm 0.3$ | 512 |
| Sparsemax | $40.0 \pm 0.2$ | $52.0 \pm 0.2$ | $46 \pm 0.3$ |
| Entmax-1.5 | $38.4 \pm 0.3$ | $49.2 \pm 0.1$ | $90 \pm 0.3$ |
| Post-hoc evidential | $38.2 \pm 0.2$ | $48.4 \pm 0.3$ | $102 \pm 0.2$ |
| **Ev-softmax** | $\mathbf{43.6 \pm 0.3}$ | $\mathbf{55.6 \pm 0.2}$ | $77 \pm 0.2$ |

which we sample. We consider a latent space of $16 \times 16$ discrete latent variables with $K = 512$ classes each. We use the same PixelCNN [30] as Itkina *et al.* [10] for the autoregressive prior. For each normalization function, we train a PixelCNN prior using that function to generate the output probability over the latent space. To evaluate the accuracy of our VQ-VAE models, we train a Wide Residual Network classifier [31] on *tiny*ImageNet and calculate its classification accuracy on the images generated by the VQ-VAE models. Further experimental details are given in Appendix E.

**Results and discussion:** Table 3 shows that *ev-softmax* outperforms other sparse methods in accuracy and is able to generate images with the most resemblance to the ground truth. Our method reduces the number of latent classes by 85% on average, more than the 83% of *entmax*-1.5 but less than the 91% of *sparsemax*. With the exception of *post-hoc evidential sparsification*, the sparse methods all outperform *softmax*, suggesting that sparsity improves the model's performance on the task of image generation by discarding irrelevant latent classes. The underperformance of the *post-hoc evidential* model is expected, as *post-hoc evidential sparsification* is applied only at test time to a model trained with the *softmax* function, which bounds its performance. Since the VQ-VAE samples directly from the output distribution of the PixelCNN prior, the probabilistic nature of this downstream task

Table 4: BLEU, ROUGE, and METEOR scores, and number of nonzero elements in the attention on the EN→DE corpus of IWSLT 2014. We report the mean and standard error for the BLEU score and number of nonzero elements over 5 random seeds. Our method outperforms all baselines in all metrics and maintains a higher number of nonzero attended elements than other sparse normalization functions. Higher is better for all metrics. The asterisk * denotes that ev-softmax outperformed the method with a significance level of 0.05, and ** denotes a significance of 0.01.

| Metric | Softmax | Sparsemax | Entmax-1.5 | Post-hoc Evidential | Ev-softmax |
|---|---|---|---|---|---|
| BLEU | $29.2 \pm 0.06$ | $29.0 \pm 0.05$ | $29.2 \pm 0.07$ | $29.2 \pm 0.05$ | $\mathbf{29.4 \pm 0.05}$ |
| ROUGE-1 | 59.31 | 58.47** | 58.94* | 59.09* | **59.32** |
| ROUGE-2 | 35.62 | 34.76** | 35.20* | 35.42* | **35.74** |
| ROUGE-L | 56.09 | 55.39** | 55.75* | 55.93* | **56.18** |
| METEOR | 57.02* | 56.33** | 5.83** | 56.84** | **57.20** |
| # attended | $39.5 \pm 11.5$ | $2.3 \pm 0.54$ | $4.1 \pm 1.3$ | $3.8 \pm 0.93$ | $8.2 \pm 1.3$ |

suggests that optimizing the prior for log-likelihood allows the model to learn a prior distribution that achieves better performance on the downstream task.

## 4.4 Machine Translation

We evaluate our method in the setting of machine translation. Attention-based translation models generally employ dense attention weights over the entire source sequence while traditional statistical machine translation systems are usually based on a lattice of sparse phrase hypotheses [9]. Thus, sparse attention offers the possibility of more interpretable attention distributions which avoid wasting probability mass on irrelevant source words. Furthermore, language modeling often requires the context of multiple words to correctly identify dependencies within an input sequence [2]. Hence, we investigate the impact of the multimodality of our proposed normalization function on the quality of translations.

**Model:** We train transformer models for each of the sparse normalization functions using the OpenNMT Neural Machine Translation Toolkit [32]. Each normalization function is used in the self-attention layers. We use byte pair encoding (BPE) to encode rare and unknown words and ensure an open vocabulary [33]. Due to computational constraints, we finetune trans-

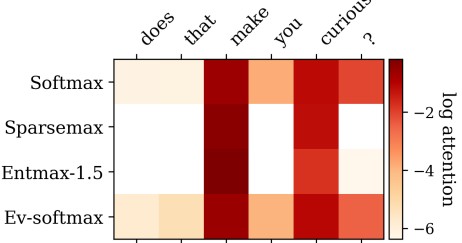

Figure 4: Attention weights produced by each normalization method for the first word of the translated sentence. *Sparsemax* and *entmax*-1.5 both attend over two words while *ev-softmax* and *softmax* attend over the entire source. The resulting predictions are *macht* (*ev-softmax*), *Ist* (*sparsemax*), and *Machen* (*entmax*-1.5, *softmax*). The ground truth translation is *macht sie das neugierig?*. Although the attention distributions of *ev-softmax* and *softmax* are similar in this example, *ev-softmax* still significantly sparsifies attention away from superfluous inputs.

former models on the English-German translation dataset from IWSLT 2014 [34]. These models were pretrained on the EN-DE corpus from WMT 2016 [35]. We finetune for 40000 iterations. Hyperparameters and further experimental details are given in Appendix E.

**Results and discussion:** As shown in Table 4, our method obtains the highest BLEU [36], METEOR [37], and ROUGE [38] scores of the considered normalization functions while reducing the average number of words attended to 8.2. In comparison with the other sparse normalization functions, ours maintains the highest number of attended source words. This supports the notion that attending over a subset of the source words benefits performance, but the performance of *sparsemax* and *entmax*-1.5 could be hurt by attending over too few words. For short sentences, *ev-softmax* maintains nonzero attention over the entire sentence, as Fig. 4 demonstrates. This allows for the model to identify relationships between words in the entire source sequence, such as tense and conjugation, enabling more accurate translation.

## 5 Related Work

**Softmax Alternatives**: There is considerable interest in alternatives to the *softmax* function that provide greater representational capacity [39], [40] and improved discrimination of features [41]. *Sparse-*

*max* [9], its generalization $\alpha$-*entmax* [16], and its *maximum a posteriori* adaptation SparseMAP [42] introduce sparse normalization functions and corresponding loss functions. While the $\alpha$-*entmax* family of functions contains a parameter $\alpha$ to control the level of sparsity, the functions are all sparse for $\alpha > 1$ with no known continuous approximation with full support. Therefore, these methods are all trained with alternative loss functions that do not directly represent the likelihood of the data. Another controllable sparse alternative to *softmax* is *sparsehourglass* [17]. We baseline against only *sparsemax* and *entmax*-1.5 as more research has been conducted on these two methods and *entmax*-1.5 is semantically similar to *sparsehourglass*. In contrast, we develop a normalization function which has a continuous family of approximations that can be used during training time with the negative log-likelihood and KL divergence loss terms. This flexibility enables our method to substitute the *softmax* transformation in any context and model.

**Discrete Variational Autoencoders**: Recent advancements in VAEs with discrete latent spaces have introduced strategies for learning discrete latent spaces to tackle a wide-ranging set of tasks including image generation [1], [43], video prediction [44], and speech processing [45]. For discrete VAEs, the gradient of the ELBO usually cannot be calculated directly [14]. While the SFE [12], [13] enables Monte Carlo-based approaches to estimate the stochastic gradient of the ELBO, it suffers from high variance and is consequently slow to converge. Continuous relaxation via the Gumbel-Softmax distribution allows for an efficient and convenient gradient calculation through discrete latent spaces [14], [15], but the gradient estimation is biased and requires sampling during training. Our work, in contrast, enables direct marginalization over a categorical space without sampling. Similar to our work, Correia *et al.* [9] apply *sparsemax* to marginalize discrete VAEs efficiently. While we also generate a sparse distribution over the latent space as in Correia *et al.* [9], we better balance sparsity with the multimodality of the distribution. Prior work on evidential sparsification focused on a post-hoc method for discrete CVAEs [10]. We extend this method to more classes of generative models and develop a modification to apply the method at training time.

**Sparse Attention**: In machine translation, sparse attention reflects the intuition that only a few source words are expected to be relevant for each translated word. The global attention model considers all words on the source side for each target word, which is potentially prohibitively expensive for translating longer sequences [46]. The *softmax* transformation has been replaced with sparse normalization functions, such as the *sparsemax* operation [11] and its generalization, $\alpha$-*entmax* [47]. Like our work, both *sparsemax* and $\alpha$-*entmax* can substitute *softmax* in the attention layers of machine translation models. Prior work on sparse normalization functions for attention layers has focused on global attention layers in recurrent neural network models. In contrast, we study the effect of incorporating these sparse normalization functions in transformers, where they are used in self-attention layers.

# 6 Conclusion

We present *ev-softmax*, a novel method to train deep generative models with sparse probability distributions. We show that our method satisfies many of the same theoretical properties as softmax and other sparse alternatives to softmax. By balancing the goals of multimodality and sparsity, our method yields better distributional accuracy and classification performance than baseline methods. Future work can involve training large computer vision and machine translation models from scratch with our method and examining its efficacy at a larger scale on more complex downstream tasks.

# 7 Broader Impact

Our work focuses on a method to train generative models that output sparse probability distributions. We apply our method in the domains of image generation and machine translation. Our work has the potential to improve the explainability and introspection of generative models. Furthermore, our work aims to reduce the computational cost of training generative models, which are used in unsupervised and semi-supervised approaches to learning. Although our empirical validation shows that our method can improve the accuracy of generated distributions, the use of sparse probability distributions can also amplify any inherent bias present in the data or modeling assumptions. One limitation of our work is that we do not provide explicit theoretical guarantees on the conditions for yielding an output probability of zero for a class. We hope that our work motivates further research into improving the accuracy of generative modeling and reducing bias of both models and data.

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
