# A  Post-Hoc Evidential Sparsification

## A.1  Evidential Theory and Logistic Regression

In evidential theory, also known as Dempster-Shafer theory [22], a *mass function* on set $\mathcal{Z}$ is a mapping $m : 2^{\mathcal{Z}} \to [0, 1]$ such that $m(\emptyset) = 0$ and,

$$\sum_{\mathcal{A} \subseteq \mathcal{Z}} m(\mathcal{A}) = 1. \tag{13}$$

Two mass functions $m_1$ and $m_2$ representing independent items of evidence can be combined using Dempster's rule [22] as,

$$(m_1 \oplus m_2)(\mathcal{A}) := \frac{1}{1 - \kappa} \sum_{\mathcal{B} \cap \mathcal{C} = \mathcal{A}} m_1(\mathcal{B}) m_2(\mathcal{C}), \tag{14}$$

for all $\mathcal{A} \subseteq \mathcal{Z}, \mathcal{A} \neq \emptyset$, and $(m_1 \oplus m_2)(\emptyset) := 0$, where $\kappa$ is the *degree of conflict* between the two mass functions, defined as,

$$\kappa := \sum_{\mathcal{B} \cap \mathcal{C} = \emptyset} m_1(\mathcal{B}) m_2(\mathcal{C}). \tag{15}$$

Classifiers that transform a linear combination of features through the softmax function can be formulated as evidential classifiers as follows [23]. Suppose $\phi \in \mathbb{R}^J$ is a feature vector and $\mathcal{Z}$ is the set of classes, with $|\mathcal{Z}| = K$, $z_k \in \mathcal{Z}$ for $k \in \{1, \ldots, K\}$. For each $z_k$, the evidence of feature $\phi_j$ is assumed to point either to the singleton $\{z_k\}$ or to its complement $\overline{\{z_k\}}$, depending on the sign of

$$w_{jk} := \beta_{jk}\phi_j + \alpha_{jk}, \tag{16}$$

where $(\beta_{jk}, \alpha_{jk})$ are parameters [23]. Then Denoeux [23] uses these weights to write mass functions

$$m_{kj}^+(\{z_k\}) = 1 - e^{-w_{jk}^+}, \quad m_{kj}^+(\mathcal{Z}) = e^{-w_{jk}^+} \tag{17}$$

$$m_{kj}^-(\overline{\{z_k\}}) = 1 - e^{-w_{jk}^-}, \quad m_{kj}^-(\mathcal{Z}) = e^{-w_{jk}^-}. \tag{18}$$

These masses can be fused through Dempster's rule to arrive at the mass function at the output of the softmax layer as follows,

$$m(\{z_k\}) \propto e^{-w_k^-} \left( e^{w_k^+} - 1 + \prod_{\ell \neq k} \left( 1 - e^{-w_\ell^-} \right) \right) \tag{19}$$

$$m(\mathcal{A}) \propto \left( \prod_{z_k \neq \mathcal{A}} \left( 1 - e^{-w_k^-} \right) \right) \left( \prod_{z_k \in \mathcal{A}} e^{-w_k^-} \right), \tag{20}$$

where $\mathcal{A} \subseteq \mathcal{Z}, |\mathcal{A}| > 1, w_k^- = \sum_{j=1}^J w_{jk}^-$, and $w_k^+ = \sum_{j=1}^J w_{jk}^+$.

## A.2  Evidential Sparsification

If no evidence directly supports class $k$ and there is no evidence contradicting another class $\ell$, then the belief mass for the singleton set $\{z_k\}$ is zero [10]. That is, if $w_k^+ = 0$ and $w_\ell^- = 0$ for at least one other class $\ell \neq k$, then $m(\{z_k\}) = 0$.

Given a neural network, Itkina *et al.* [10] construct evidential weights from the output of the last hidden layer in a neural network $\phi \in \mathbb{R}^J$ and output linear layer weights $\hat{\beta} \in \mathbb{R}^{J \times K}$ as follows:

$$w_{jk} = \beta_{jk}\phi_j + \alpha_{jk}, \tag{21}$$

where $\beta_{jk} = \hat{\beta}_{jk} - \frac{1}{K}\sum_{\ell=1}^K \hat{\beta}_{j\ell}$ and $\alpha_{jk} = \frac{1}{J}\left(\beta_{0k} + \sum_{j=1}^J \beta_{jk}\phi_j\right) - \beta_{jk}\phi_j$. These evidential weights $w_{jk}$ do not depend on $j$ (i.e. $w_{0k} = w_{1k} = \ldots = w_{Jk}$), which implies that $w_k^+ = \max(0, w_k)$ and $w_k^- = \max(0, -w_k)$. The singleton mass function (Eq. (19)) obtained by fusing these weights is then used to identify latent classes to be filtered as follows:

$$p_{\text{filtered}}(z_k|\phi) \propto \mathbb{1}\{m(\{z_k\}) \neq 0\} p_{\text{softmax}}(z_k|\phi), \tag{22}$$

where $p_{\text{softmax}}(z_k|\phi)$ is the $k$th element of the distribution obtained from applying the softmax transformation to the vector $\hat{\beta}^T \phi$.

# B Equivalence to Post-Hoc Evidential Sparsification Function

In this section, we derive equivalence of the evidential softmax (*ev-softmax*) function and the post-hoc sparsification function introduced by Itkina *et al.* [10]. We begin with two definitions of the post-hoc evidential sparsification method.

**Definition B.1.** *Given a feature vector $\phi \in \mathbb{R}^J$, weights $\hat{\beta} \in \mathbb{R}^{J \times K}$, and a bias vector $\hat{\alpha} \in \mathbb{R}^K$, the **evidential weights matrix** $W \in \mathbb{R}^{J \times K}$ is defined element-wise as*

$$w_{jk} = \frac{1}{J}\left(\alpha_k + \sum_{j=1}^{J} \beta_{jk}\phi_j\right), \tag{23}$$

*where $\beta_{jk} = \hat{\beta}_{jk} - \frac{1}{K}\sum_{\ell=1}^{K} \hat{\beta}_{j\ell}$ and $\alpha_k = \hat{\alpha}_k - \frac{1}{K}\sum_{\ell=1}^{K} \hat{\alpha}_\ell$ are normalized weights and bias terms, respectively.*

**Definition B.2.** *Define the **evidential class weights** $w, w^+, w^- \in \mathbb{R}^K$ element-wise as*

$$w_k = \sum_{j=1}^{J} w_{jk} = \alpha_k + \sum_{j=1}^{J} \beta_{jk}\phi_j \tag{24}$$

$$w_k^+ = \sum_{j=1}^{J} \max(0, w_{jk}) = \max\left(0, \alpha_k + \sum_{j=1}^{J} \beta_{jk}\phi_j\right) = \max(0, w_k) \tag{25}$$

$$w_k^- = \sum_{j=1}^{J} \max(0, -w_{jk}) = \max\left(0, -\alpha_k - \sum_{j=1}^{J} \beta_{jk}\phi_j\right) = \max(0, -w_k). \tag{26}$$

*In matrix notation, we write this as $w^+ = \max(0, \alpha + \beta^T\phi)$, $w^- = \max(0, -\alpha - \beta^T\phi)$, where the* max *operator is applied at each index independently. Note this is the same formulation as defined in Appendix A.*

We prove a lemma showing an equivalent definition that involves centering the output class weights $w$ instead of centering the input weights $\hat{\alpha}, \hat{\beta}$.

**Lemma B.1.** *For $\phi \in \mathbb{R}^J, \hat{\beta} \in \mathbb{R}^{J \times K}, \hat{\alpha} \in \mathbb{R}^K$, and $w \in \mathbb{R}^K$ as defined in Definition B.1,*

$$w_k = \hat{\alpha}_k + \sum_{j=1}^{J} \hat{\beta}_{jk}\phi_j - \frac{1}{K}\sum_{\ell=1}^{K}\left(\hat{\alpha}_\ell + \sum_{m=1}^{J} \hat{\beta}_{m\ell}\phi_m\right). \tag{27}$$

*Proof.* We substitute the definitions of $\alpha$ and $\beta$ into Eq. (23) to obtain

$$
\begin{aligned}
w_k &= \alpha_k + \sum_{j=1}^{J} \beta_{jk}\phi_j \\
&= \left(\hat{\alpha}_k - \frac{1}{K}\sum_{\ell=1}^{K} \hat{\alpha}_\ell\right) + \sum_{j=1}^{J}\left(\hat{\beta}_{jk} - \frac{1}{K}\sum_{\ell=1}^{K} \hat{\beta}_{j\ell}\right)\phi_j \\
&= \hat{\alpha}_k - \frac{1}{K}\sum_{\ell=1}^{K} \hat{\alpha}_\ell + \sum_{j=1}^{J}\left(\hat{\beta}_{jk}\right) - \frac{1}{K}\sum_{\ell=1}^{K}\sum_{m=1}^{J} \hat{\beta}_{m\ell}\phi_m \\
&= \hat{\alpha}_k + \sum_{j=1}^{J} \hat{\beta}_{jk}\phi_j - \frac{1}{K}\sum_{\ell=1}^{K}\left(\hat{\alpha}_\ell + \sum_{m=1}^{J} \hat{\beta}_{m\ell}\phi_m\right).
\end{aligned}
\tag{28}
$$

$\square$

Note as a consequence that the sum of the evidential weights is zero. That is,

$$
\begin{aligned}
\sum_{k=1}^{K} w_k &= \sum_{k=1}^{K} \left( \hat{\alpha}_k + \sum_{j=1}^{J} \hat{\beta}_{jk} \phi_j - \frac{1}{K} \sum_{\ell=1}^{K} \left( \hat{\alpha}_\ell + \sum_{m=1}^{J} \hat{\beta}_{m\ell} \phi_m \right) \right) \\
&= \sum_{k=1}^{K} \left( \hat{\alpha}_k + \sum_{j=1}^{J} \hat{\beta}_{jk} \phi_j \right) - \sum_{\ell=1}^{K} \left( \hat{\alpha}_\ell + \sum_{m=1}^{J} \hat{\beta}_{m\ell} \phi_m \right) \\
&= 0.
\end{aligned}
\tag{29}
$$

We define the **mass function** $m$ and **post-hoc evidential sparsification function** $f$ as in Eqs. (19) and (22). Next we prove a key lemma, which is a condition for determining which terms in the probability function $f$ are set to 0 in the post-hoc evidential sparsification function.

**Lemma B.2.** *For fixed $k \in \{1, \ldots, K\}$ and $\boldsymbol{w}, m$ as defined in Definition B.2 and Eq. (19), $m\{z_k\} = 0$ if and only if $w_k \leq 0$.*

*Proof.* Take fixed $k \in \{1, \ldots, K\}$. For both directions, we rely on the key observation that $m(\{z_k\}) = 0$ if and only if the following two conditions are both true [10]:

$$
w_k^+ = 0 \tag{30}
$$

$$
w_\ell^- = 0 \text{ for some } \ell \neq k. \tag{31}
$$

We first prove the forward direction, that $m(\{z_k\}) = 0$ implies $w_k \leq 0$. Using the observation above, $m(\{z_k\})$ implies that $w_k^+ = 0$. Since $w_k^+ = \max(0, w_k)$ by definition, we see that $w_k \leq 0$ as desired.

To prove the reverse, note that $w_k \leq 0$ implies that $w_k^+ = \max(0, w_k) = 0$. Now since the evidential class weights have a sum of 0 (Eq. (29)). If $w_k \leq 0$, then either $\boldsymbol{w}$ is the zero vector or $\boldsymbol{w}$ must contain some positive element $w_\ell > 0$, where $\ell \neq k$. In either case, there must exist $\ell \neq k$ such that $w_\ell^- = 0$. Hence, $w_k^+ = 0$ and the existance of $\ell \neq k$ such that $w_\ell^- = 0$ implies that $m(\{z_k\}) = 0$ as desired. $\square$

Lemma B.2 leads to the following natural definition for the *ev-softmax* function EVSOFTMAX$'$.

**Definition B.3.** *Given a vector $\hat{\boldsymbol{v}} \in \mathbb{R}^K$, define the **evidential softmax function** EVSOFTMAX$'$ : $\mathbb{R}^K \rightarrow \Delta^K$ as*

$$
\text{EVSOFTMAX}'(\hat{\boldsymbol{v}})_k = \begin{cases} \frac{1}{K} & \text{if } \boldsymbol{v} = \boldsymbol{0} \\ \frac{\mathbb{1}\{v_k > 0\} e^{v_k}}{\sum_{\ell=1}^{K} \mathbb{1}\{v_k > 0\} e^{v_\ell}} & \text{otherwise} \end{cases}
\tag{32}
$$

*where $\boldsymbol{v} = \hat{\boldsymbol{v}} - \frac{1}{K} \sum_{k=1}^{K} \hat{v}_k$ is centered to have a mean of 0.*

*We now define a function which is equivalent under certain conditions. If the marginal probability measure of $v_k$ is non-atomic for each $k$, then the following function is equal to Eq. (32) with probability 1:*

$$
\text{EVSOFTMAX}(\boldsymbol{v}) = \frac{\mathbb{1}\{v_k \geq 0\} e^{v_k}}{\sum_{\ell=1}^{K} \mathbb{1}\{v_k \geq 0\} e^{v_\ell}}.
\tag{33}
$$

EVSOFTMAX*' is nearly equivalent to* EVSOFTMAX*, with the difference lying in the equality condition in the indicator function.*

We formalize the equivalence of Eq. (32) and Eq. (33) with the following lemma.

**Lemma B.3.** *With $\hat{\boldsymbol{v}}$ and $\boldsymbol{v}$ defined as in Definition B.3, if the marginal probability measure of $v_k$ is non-atomic for each $k$, then Equations Eq. (32) and Eq. (33) are equal with probability 1.*

*Proof.* We can see that the expressions for EVSOFTMAX$(\mathbf{v})$ and EVSOFTMAX$'(\mathbf{v})$ are equal for all $\mathbf{v}$ such that $v_k \neq 0$ for all $k$. Now we can apply the union bound to this event as follows:

$$
P(\cap_k \{v_k \neq 0\}) = 1 - P(\cup_k \{v_k = 0\}) \geq 1 - \sum_k P(\{v_k = 0\}) = 1
$$

where the last equality follows from the assumption that each $v_k$ has a non-atomic distribution. $\square$

We are now ready to prove the main result.

**Theorem B.4.** *Given a feature vector $\phi \in \mathbb{R}^J$, weights $\hat{\beta} \in \mathbb{R}^{J \times K}$, and bias vector $\hat{\alpha} \in \mathbb{R}^K$, define the **evidential class weights** $\boldsymbol{w}, \boldsymbol{w}^+, \boldsymbol{w}^- \in \mathbb{R}^K$ as in Definition B.1, the **post-hoc evidential sparsification function** $f : \mathbb{R}^K \to \Delta^K$ as in Eq. (22), and EVSOFTMAX as in Eq. (33).*

*If the marginal distributions of the weights $w_k$ are non-atomic for all $k \in \{1, \dots, K\}$, then the following equality holds with probability 1:*

$$f(\boldsymbol{w}) = \text{EVSOFTMAX}(\hat{\beta}^T \phi + \hat{\alpha}). \tag{34}$$

*Proof.* Let $\hat{\mathbf{v}} = \hat{\beta}^T \phi + \hat{\alpha}$. Then Lemma B.1 shows that $\boldsymbol{w}$ is equivalent to the normalization of $\hat{\mathbf{v}}$ (e.g. $w_k = \hat{v}_k - \frac{1}{K} \sum_{j=1}^{K} \hat{v}_j$), hence we have

$$\text{EVSOFTMAX}(\hat{\beta}^T \phi + \hat{\alpha})_k = \frac{\mathbb{1}\{w_k \geq 0\} e^{w_k}}{\sum_{\ell=1}^{K} \mathbb{1}\{w_k \geq 0\} e^{w_\ell}}. \tag{35}$$

Now Lemma B.2 and Lemma B.3 combined with the assumption that the marginal distributions of $w_k$ are non-atomic imply that $\mathbb{1}\{m\{z_k\} \neq 0\} = \mathbb{1}\{w_k \geq 0\}$ for all $k \in \{1, \dots, K\}$ with probability 1, and the result follows. $\square$

Thus we show that the *ev-softmax* function of Eq. (33) is equivalent to the post-hoc evidential sparsification function detailed in Appendix A.

## C  Properties of Evidential Softmax

We prove the following properties of the *ev-softmax* transformation (Eq. (7)):

1. **Monotonicity**: If $v_i \geq v_j$, then $\text{EVSOFTMAX}(\mathbf{v})_i \geq \text{EVSOFTMAX}(\mathbf{v})_j$.

   *Proof.* If $v_i \geq v_j$ then $\mathbb{1}\{v_i \geq 0\} \geq \mathbb{1}\{v_j \geq 0\} \geq 0$ and $e^{v_i} \geq e^{v_j} \geq 0$. Multiplying the two inequalities gives the desired result. $\square$

2. **Full domain**: $\text{dom}(\text{EVSOFTMAX}) = \mathbb{R}^K$.

   *Proof.* Since the input vector is normalized, $\sum_{k=1}^{K} \mathbb{1}\{v_k \geq \bar{v}\} e^{v_k}$ is guaranteed to be positive, so the function is defined for all $\boldsymbol{w} \in \mathbb{R}^K$ and always maps onto the simplex. $\square$

3. **Existence of Jacobian**: The Jacobian is defined for all $\mathbf{v} \in \mathbb{R}^K$ such that $v_k \neq \frac{1}{K} \sum_j v_j$ for all $k$, and it has the form

   $$\frac{\partial \text{EVSOFTMAX}(\mathbf{v})_i}{\partial v_j} = \frac{\mathbb{1}\{\hat{v}_i \geq 0\} \mathbb{1}\{\hat{v}_j \geq 0\} \left( \delta_{ij} e^{\hat{v}_i} \sum_k \mathbb{1}\{\hat{v}_k \geq 0\} e^{\hat{v}_k} - e^{\hat{v}_i} e^{\hat{v}_j} \right)}{\left( \sum_k \mathbb{1}\{\hat{v}_k \geq 0\} e^{\hat{v}_k} \right)^2} \tag{36}$$

   $$= \text{EVSOFTMAX}(\mathbf{v})_i (\delta_{ij} - \text{EVSOFTMAX}(\mathbf{v})_j) \tag{37}$$

   where $\hat{\mathbf{v}} = v_k - \frac{1}{K} \sum_k v_k$.
   Furthermore, if the marginal probability measure of $\hat{v}_k$ is non-atomic for each $k$, then the Jacobian is defined with probability 1.

   *Proof.* Let $A := \{k \mid \hat{v}_k \geq 0\}$. Then equivalently we can write Eq. (36) as

   $$\frac{\partial \text{EVSOFTMAX}(\mathbf{v})_i}{\partial v_j} = \begin{cases} 0 & \text{if } i \notin A \text{ or } j \notin A \\ \frac{\delta_{ij} e^{\hat{v}_i} \sum_{k \in A} e^{\hat{v}_k} - e^{\hat{v}_i} e^{\hat{v}_j}}{\left( \sum_{k \in A} e^{v_k} \right)^2} & \text{otherwise.} \end{cases} \tag{38}$$

   Take arbitrary $i \notin A$, then $\text{EVSOFTMAX}(\mathbf{v})_i = 0$, and since by assumption $\hat{v}_k \neq 0$ for all $k$, there must exist some $\epsilon > 0$ such that we have $\text{EVSOFTMAX}(\mathbf{v}')_i = 0$ for all $\mathbf{v}'$ within a radius of $\epsilon$ from $\mathbf{v}$. This implies that $\frac{\partial \text{EVSOFTMAX}(\mathbf{v})_i}{\partial v_j} = 0$ for all $j \in \{1, \dots, K\}$.

Now take arbitrary $i \in A$ and $j \notin A$. Then by assumption, $\hat{v}_j < 0$ which implies that there exists $\epsilon > 0$ such that for all $v'_j$ within $\epsilon$ of $v_j$, we have $\mathbb{1}\{v'_j \geq 0\} = 0$. This means that EVSOFTMAX$(\mathbf{v})_i$ is independent of $v'_j$ in this neighborhood, giving $\frac{\partial \text{EVSOFTMAX}(\mathbf{v})_i}{\partial v_j} = 0$ as desired for all $i \in \{1, \ldots, K\}$.

We prove the final case, where both $i, j \in A$. In this case, the expression for $g_i(\mathbf{v})$ is exactly that of softmax over variables in $A$, so the Jacobian over all variables $j \in A$ must also match that of softmax (i.e. $\frac{\partial \text{SOFTMAX}(\mathbf{v})_i}{\partial v_j} = \text{SOFTMAX}(\mathbf{v})_i(\delta_{ij} - \text{SOFTMAX}(\mathbf{v})_j)$), and Eq. (38) follows.

Next, we show the equivalence of Eq. (38) to Eq. (37). If $i \notin A$ or $j \notin A$, we can check by inspection that EVSOFTMAX$(\mathbf{v})_i(\delta_{ij} - \text{EVSOFTMAX}(\mathbf{v})_j) = 0$. Otherwise, EVSOFTMAX simply reduces to softmax over the indices in $A$, and the result follows from the analogous equation for softmax (i.e. $\frac{\partial \text{SOFTMAX}(\mathbf{v})_i}{\partial v_j} = \text{SOFTMAX}(\mathbf{v})_i(\delta_{ij} - \text{SOFTMAX}(\mathbf{v})_j)$).

Finally, we show that the Jacobian is defined with probability 1, assuming that the probability measure of $\hat{v}_k$ for each $k$ is nonatomic. The above shows the Jacobian is defined for all $\mathbf{v}$ such that $\hat{v}_k \neq 0$ for all $k$. Observe by the union bound that $P(\cap_k \{\hat{v}_k \neq 0\}) \geq 1 - \sum_k P(\{\hat{v}_k = 0\})$. For each $k$, the measure of $\hat{v}_k$ is non-atomic, so $P(\{\hat{v}_k = 0\}) = 0$ and the result follows. $\qquad\square$

4. **Lipschitz continuity**: There exists $L \geq 0$ such that for any $\mathbf{v}_1, \mathbf{v}_2, \|\text{EVSOFTMAX}(\mathbf{v}_1) - \text{EVSOFTMAX}(\mathbf{v}_2)\|_2 \leq L\|\mathbf{v}_1 - \mathbf{v}_2\|_2$. The evidential softmax function is Lipschitz with Lipschitz constant 1 provided the two outputs EVSOFTMAX$(\mathbf{v}_1)$, EVSOFTMAX$(\mathbf{v}_2)$ have the same support.

   *Proof.* This follows from the Lipschitz continuity of the softmax function with Lipschitz constant 1 [17]. $\qquad\square$

5. **Translation invariance**: Adding a constant to evidential softmax does not change the output since the input is normalized around its mean.

   *Proof.* This follows since all input vectors $v$ are normalized by their mean $\frac{1}{K}\sum_{j=1}^{K} v_j$. $\qquad\square$

6. **Permutation invariance**: Like softmax, evidential softmax is permutation invariant.

   *Proof.* This follows from the coordinate-symmetry in the equations in Eq. (7). $\qquad\square$

# D   Gradient of the Evidential Softmax Loss

In this section, we prove Eq. (12), which amounts to the claim that the gradient of the log likelihood of EVSOFTMAX$_{\text{train},\epsilon}$ (Eq. (10)) approaches the same form as that of softmax as $\epsilon$ approaches 0.

*Proof.* The existence of the Jacobian in Eq. (37) gives,

$$\frac{\partial \text{EVSOFTMAX}(\mathbf{v})_i}{\partial v_j} = \text{EVSOFTMAX}(\mathbf{v})_i(\delta_{ij} - \text{EVSOFTMAX}(\mathbf{v})_j) \qquad (39)$$

for each input $\mathbf{v}$ and each $i, j$. Now by the continuity of the exponential function, and therefore the continuity of the EVSOFTMAX function, it follows that

$$\lim_{\epsilon \to 0} \text{EVSOFTMAX}_{\text{train},\epsilon}(\mathbf{v})_i = \text{EVSOFTMAX}(\mathbf{v})_i$$
$$\lim_{\epsilon \to 0} \delta_{ij} - \text{EVSOFTMAX}_{\text{train},\epsilon}(\mathbf{v})_j = \delta_{ij} - \text{EVSOFTMAX}(\mathbf{v})_j$$
$$\lim_{\epsilon \to 0} \frac{\partial \text{EVSOFTMAX}_{\text{train},\epsilon}(\mathbf{v})_i}{\partial v_j} = \frac{\partial \text{EVSOFTMAX}(\mathbf{v})_i}{\partial v_j}. \qquad (40)$$

Therefore,

$$
\begin{aligned}
\lim_{\epsilon \to 0} \frac{\partial \text{EvSoftmax}_{\text{train},\epsilon}(\mathbf{v})_i}{\partial v_j} &= \lim_{\epsilon \to 0} \text{EvSoftmax}_{\text{train},\epsilon}(\mathbf{v})_i (\delta_{ij} - \text{EvSoftmax}_{\text{train},\epsilon}(\mathbf{v})_j) \\
&= \text{EvSoftmax}(\mathbf{v})_i (\delta_{ij} - \text{EvSoftmax}(\mathbf{v})_j).
\end{aligned} \tag{41}
$$

To compute the gradient, we use the chain rule:

$$
\begin{aligned}
&\lim_{\epsilon \to 0} \frac{\partial}{\partial v_j} \log \text{EvSoftmax}_{\text{train},\epsilon}(\mathbf{v})_i \\
&= \lim_{\epsilon \to 0} \frac{1}{\text{EvSoftmax}_{\text{train},\epsilon}(\mathbf{v})_i} \frac{\partial \text{EvSoftmax}_{\text{train},\epsilon}(\mathbf{v})_i}{\partial v_j} \\
&= \lim_{\epsilon \to 0} \frac{1}{\text{EvSoftmax}_{\text{train},\epsilon}(\mathbf{v})_i} \lim_{\epsilon \to 0} \frac{\partial \text{EvSoftmax}_{\text{train},\epsilon}(\mathbf{v})_i}{\partial v_j} \\
&= \lim_{\epsilon \to 0} \frac{1}{\text{EvSoftmax}_{\text{train},\epsilon}(\mathbf{v})_i} \lim_{\epsilon \to 0} \text{EvSoftmax}_{\text{train},\epsilon}(\mathbf{v})_i (\delta_{ij} - \text{EvSoftmax}_{\text{train},\epsilon}(\mathbf{v})_j) \\
&= \lim_{\epsilon \to 0} \frac{1}{\text{EvSoftmax}_{\text{train},\epsilon}(\mathbf{v})_i} \text{EvSoftmax}_{\text{train},\epsilon}(\mathbf{v})_i (\delta_{ij} - \text{EvSoftmax}_{\text{train},\epsilon}(\mathbf{v})_j) \\
&= \lim_{\epsilon \to 0} \delta_{ij} - \text{EvSoftmax}_{\text{train},\epsilon}(\mathbf{v})_j \\
&= \delta_{ij} - \text{EvSoftmax}(\mathbf{v})_j.
\end{aligned} \tag{42}
$$

$\square$

## E  Further Experimental Details

All experiments were performed on a single local NVIDIA GeForce GTX 1070 or Tesla K40c GPU.

### E.1  MNIST CVAE

We train a CVAE architecture [19] with two multi-layer perceptrons (MLPs) for the encoder and one MLP for the decoder. For the MLPs of the encoder and decoder, we use two fully connected layers per MLP. For the encoder, we use hidden unit dimensionalities of 30 for $p_\theta(\mathbf{z} \mid y)$ and 256 for $q_\phi(\mathbf{z} \mid \mathbf{x}, y)$, and for the decoder, we use hidden unit dimensionality of 256 for $p(\mathbf{x}' \mid \mathbf{z})$. We use the ReLU nonlinearity with stochastic gradient descent and a learning rate of 0.001. During training time, the Gumbel-Softmax reparameterization was used to backpropagate loss gradients through the discrete latent space [14], [15]. We train for 20 epochs with a batch size of 64. The standard conditional evidence lower bound (ELBO) of Eq. (2) was maximized to train the model.

For each normalization function $f$ (*softmax*, *sparsemax*, *entmax*-1.5, *post-hoc evidential*, and *ev-softmax*), we use the corresponding decoder to generate one image $\hat{x}_k^{(f)}$ for each latent class $z_k^{(f)} \in \{z_0^{(f)}, \ldots, z_9^{(f)}\}$. For each image $\hat{x}_k^{(f)}$, we use the trained classifier to calculate the probability distribution $Q(\tilde{\mathbf{z}} \mid \hat{x}_k^{(f)})$, which represents the probability distribution over the ten classes of handwritten digits inferred from the decoded image. Let $p_\theta(\mathbf{z} \mid \mathbf{y}; f)$ denote the prior distribution over the latent classes generated by normalization function $f$ with model parameterized by $\theta$. We define $p(\tilde{\mathbf{z}} \mid \mathbf{y}; f) = \sum_{k=1}^{10} Q(\tilde{\mathbf{z}} \mid \hat{x}_k^{(f)}) p_\theta(z_k \mid \mathbf{y}; f)$ as the probability distribution over the ten classes of handwritten digits marginalized over the prior distribution. Then the Wasserstein distance is calculated between $p(\tilde{\mathbf{z}} \mid y; f)$ and the uniform distribution over $\tilde{z}_1, \tilde{z}_3, \tilde{z}_5, \tilde{z}_7, \tilde{z}_9$ for $y =$ odd, and $\tilde{z}_0, \tilde{z}_2, \tilde{z}_4, \tilde{z}_6, \tilde{z}_8$ for $y =$ even.

### E.2  Semi-supervised VAE

We use a classification network consisting of three fully connected hidden layers of size 256, using ReLU activations. The generative and inference network both consist of one hidden layer of size 128 with ReLU activations. The multivariate Gaussian has 8 dimensions with a diagonal covariance matrix. We use the Adam optimizer with a learning rate of $5 \cdot 10^{-4}$. For the labeled loss component

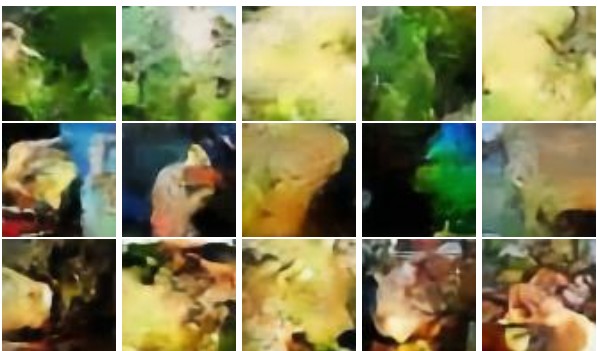

Figure 5: Generated images for the queries "cauliflower", "jellyfish", and "mashed potato" with the model using ev-softmax.

of the semi-supervised objective, we use the *sparsemax loss* and *Tsallis loss* for the models using *sparsemax* and *entmax*-1.5, respectively, and we use the negative log-likelihood loss for the models using *softmax* and *ev-softmax*. Following Liu *et al.* [21], we pretrain the network with only labeled data prior to training with the whole training set.

### E.3   VQ-VAE

We train the VQ-VAE [1] network on *tiny*ImageNet data normalized to $[-1, 1]$. The *tiny*ImageNet dataset consists of 10,000 images and 200 classes. We use *tiny*ImageNet due to its more computational feasible size for training on a single NVIDIA GeForce GTX 1070 GPU. We train the VQ-VAE with the default parameters from `https://github.com/ritheshkumar95/pytorch-vqvae`. We use a batch size of 128 for 100 epochs, $K = 512$ for the number of classes for each of the $16 \times 16$ latent variables, a hidden size of 128, and a $\beta$ of one. The network was trained with the Adam optimizer [48] with a starting learning rate of $2 \times 10^{-4}$. We then train PixelCNN [30] priors for each normalization function (*softmax*, *sparsemax*, *entmax*-1.5, *post-hoc evidential*, and *ev-softmax*) over the latent space with 10 layers, hidden dimension of 128, and batch size of 32 for 100 epochs. The networks for each function were trained with the Adam optimizer over learning rates of $10^{-5}$ and $10^{-6}$. For each normalization function, the network was chosen by selecting the one with the lowest loss on the validation set. We generate a new dataset by sampling from the trained prior, and decoding the images using the VQ-VAE decoder. We sample 25 latent encodings from the prior for each of the 200 *tiny*ImageNet training classes to build a dataset for each normalization function.

We then train Wide Residual Networks (WRN) [31] for classification on *tiny*ImageNet. Each WRN is trained for 100 epochs with a batch size of 128. The optimizers are Adam with learning rates of $10^{-4}, 10^{-5}$, and $10^{-6}$. The best WRN was selected through validation accuracy. The inference performance of the WRN classifier on the datasets generated with the *softmax*, *sparsemax*, *entmax*-1.5, and *ev-softmax* distributions are compared, demonstrating that our distribution yields the best performance while significantly reducing the size of the latent sample space.

Fig. 5 shows sampled images generated using the proposed evidential softmax normalization function. The spatial structure of the queries is demonstrated in the generated samples.

### E.4   Transformer

We train transformer models [2] on the English-German corpus of IWSLT 2014 [34]. The models were pretrained on the WMT 2016 English-German corpus [35]. We use the OpenNMT implementation [30]. For each normalization function, we use a batch size of 1024, word vector size of 512, and inner-layer dimensionality of 2048. We train with Adam with a total learning rate of 0.5 using the same learning rate decay of Vaswani *et al.* [2], and we train over 50000 iterations. We report tokenized test BLEU scores across five random seeds.