# OpenReview forum: "Evidential Softmax for Sparse Multimodal Distributions in Deep Generative Models"
_NeurIPS.cc/2021/Conference — NeurIPS 2021 Poster_

### Official Review · Reviewer_R6Na · 2021-07-11

**Rating:** 6
**Confidence:** 3

**Summary:**

This paper shows a sparse normalization function that preserves the multimodality of probability distributions to  train deep generative models.

The authors apply their methods to different VAE architectures and demonstrate the effectiveness of their methods.


**Limitations And Societal Impact:**

The authors do not fully demonstrate the limitations of their work. In addition, they do not clearly give the experimental results to show the benefits of considering the multimodality and sparsity of the data distribution.

**Main Review:**

The overall framework of the paper is good. However, the motivation of the paper is not very clear. Compared to the previous works about VAE, this paper is not very innovative.

**Time Spent Reviewing:**

1.5

---

> ### Author Response · Authors · 2021-08-10
> **Response and Clarifications to Reviewer R6Na**
>
> We thank the reviewer for the critique. We respond below to points individually:
>
> 1. > The authors apply their methods to different VAE architectures and demonstrate the effectiveness of their methods.
>
> In addition to three VAE architectures in the experiments, we also apply the methods to a transformer-based language model.
>
> 2. > However, the motivation of the paper is not very clear.
>
> We clarify the motivation in the introduction of the paper as follows. Discrete latent spaces are often high dimensional. As a result, downstream tasks, such as conditional image generation, can be intractable to perform exactly as they require marginalization over the latent space. Sparse normalization functions can alleviate this intractability by reducing the number of nonzero latent classes over which to marginalize. However, many existing sparse normalization functions collapse multimodality in distributions, which degrades the performance in tasks such as conditional image generation. The post-hoc function introduced by Itkina et al. [10] addresses these shortcomings for test time. Our proposed evidential softmax (ev-softmax) function further generalizes this post-hoc function to training-time application. We present a simplified expression of the post-hoc function in Itkina et al. [10] and an investigation of its properties.
>
> 3. > Compared to the previous works about VAE, this paper is not very innovative.
>
> The novelty of our paper is three-fold:
> - We propose a novel normalization function that is differentiable, as well as a continuous family of approximations with full support. As a result, the normalization function can be used during **both training and testing** as opposed to other approaches such as Itkina et al. [10] that can only be used **during testing**. This enables the downstream tasks to be trained with sparse inputs, potentially improving performance on a wide range of tasks.
> - To the best of our knowledge, there is no existing work that provides theoretical analysis for training sparse normalization functions with NLL and KL based losses, which are relevant for probabilistic neural network models.
> - We demonstrate the applicability of our method on a wide range of probabilistic models, from autoencoding models (VAEs and CVAEs) to an autoregressive language model (transformer).
>
> 4. > The authors do not fully demonstrate the limitations of their work.
>
> In Section 6, we state the examination of evidential softmax at a larger scale as future work. In Section 7, we also detail that we do not provide explicit theoretical guarantees on the conditions for yielding an output probability of zero for a class.
>
> 5. > In addition, they do not clearly give the experimental results to show the benefits of considering the multimodality and sparsity of the data distribution.
>
> We would like to clarify the potential misunderstanding here. We use the conditional variational autoencoder experiment to provide a visualization of how the combination of multimodality and sparsity can yield significant improvements on a conditional task with a true prior distribution that is multimodal. In Fig. 1, only evidential softmax is able to learn 5-modal distributions, given even or odd as an input. As a result, the learned latent classes using evidential softmax also match most closely to the ten digits, compared to the learned latent classes using other normalization functions. Without sparsity, softmax yields a distribution which spans all ten latent classes for both even and odd inputs. Without multimodality, sparsemax and entmax-$1.5$ collapse modes of the distribution, resulting in degenerate latent classes.

---

### Official Review · Reviewer_G99K · 2021-07-12

**Rating:** 8
**Confidence:** 4

**Summary:**

The paper introduces a normalizing function called evidential softmax (ev-softmax). Based on principles of evidential theory, ev-softmax is able to assign zero probability to classes that lack evidence of the data. It can be seen as a generalization of the work done by Itkina et al. (2020), the crucial difference being that ev-softmax can be applied at training time. The experiment section shows improved performance compared to other sparse methods, on a variety of tasks.

**Limitations And Societal Impact:**

I believe the authors have adequately addressed the societal impact of their work.

**Main Review:**

_Originality_: The submission introduces a novel and differentiable normalizing function that is a generalization of the method described in Itkina et al. (2020). It is clear how this work differs from that contribution, and related work is adequately cited, as far as I know.

_Quality_: The methods described in this submission are well supported by evidential theory and the authors do a good job in their explanation. Furthermore, they use their method in a variety of tasks, in which they empirically show that their method has superior performance when compared to sparsemax, 1.5-entmax, and softmax.

_Clarity_: This submission is very well written and well detailed. Results could be easily reproduced since the authors provided the code.

_Significance_: The method proposed in this submission can be an asset to researchers and practitioners in the future as a promising sparse alternative to softmax, where the reason for this sparsity is theoretically grounded in evidential theory. The improved results of ev-softmax on four different tasks also show that the method has practical applications.

**Time Spent Reviewing:**

4

---

> ### Author Response · Authors · 2021-08-10
> **Response and Clarifications to Reviewer G99K**
>
> We thank the reviewer for the constructive review. We are happy to see that the reviewer found ev-softmax as a promising sparse alternative to softmax, and the improved results on four different tasks as compelling for practical applications.

---

> > ### Comment · Reviewer_G99K · 2021-08-13
> > **Response to rebuttal**
> >
> > After reading the other reviews and the respective rebuttals, I decided to keep my score.

---

### Official Review · Reviewer_fAmj · 2021-07-16

**Rating:** 6
**Confidence:** 3

**Summary:**

This paper proposes a sparse normalization function called evidential softmax (ev-softmax). Specifically, it extends the post-hoc sparsification approach of Itkina et al, to propose a normalization function that can be applied during both training and testing. Notably, the authors claim that ev-softmax preserves the multimodality of the distribution, while providing sparse output distributions. The authors argue that sparsity is likely to help with interpretability and also improve task performance, while multimodality is useful when the output space is multimodal. They also show experiments showing that previous sparse normalization functions collapse multimodality of distributions.

The formulation of the ev-softmax function is accompanied with relevant proofs showing that a continuous family of approximations exist for ev-softmax, which make it possible to train with NLL and KL divergence based losses. Experiments on image generation using a generative model (VAE) and machine translation (using an autoregressive model) show some improvements over previous sparse normalization techniques on distributional accuracy and classification performance.


**Limitations And Societal Impact:**

I would appreciate if the authors could expand upon the discussion in section 7 about the possibility of bias amplification through sparsification. I also wonder if there is relevant work that can be cited to add on to this discussion.


**Main Review:**

The paper provides a clear description of the formulation of the ev-softmax operation and a comprehensive list of properties exhibited by this function. The argument for building sparse yet multimodal distributions is also convincing. However, I'm not sure I completely understand the advantage of applying evidential softmax at training time. It would be great if the authors could clarify. Further, I believe the authors could have still compared with only applying post-hoc sparsification using the method of Itkina et al just to provide a data point for comparison. I believe this could be discussed more in the paper to emphasize the novelty of the work.

One other comment I had about the description is that currently, it might not be giving enough intuition to the readers about why ev-softmax gives sparse & multimodal distributions. This might be because that this work is an extension of the work of Itkina et al and some details are redundant, but I still think it could be worth illustrating in more depth how this function behaves and why it fits the desired criteria.

On the experimental front, I think the paper could be improved. The improvements on some of the tasks (in tables 2, 4) seem very small and significance tests are missing. I also wonder if the authors could have used considered different values of alpha for the alpha-entmax baseline. Following are my questions/concerns about the experiments:
- Re: the MNIST experiments in sec 4.1, both sparsemax and entmax-1.5 seem to have degenerate performance in this experiment but they perform reasonably well in the other image generation experiments. Do you know why this is the case?
- Did you consider varying the value of alpha in entmax? If yes, does that help improve the multimodality of the distributions?
- How well does the wide residual network used for evaluating the models in the exp in 4.3 perform? Instead of using a model for evaluation, did you consider performing human evaluation on a smaller sample?
- The improvements on the machine translation experiments appear to be very small. Have you performed significance testing (perhaps using a paired bootstrap test)? Computing multiple automatic metrics such as ROUGE, METEOR and performing human evaluation would be useful too.
- How can one guarantee that it is multimodality that is helping performance? I understand the # attended words is higher / latent space reduction is (sometimes) higher with ev-softmax. However, there is lack of evidence showing that multimodality helps performance.
- How expensive is the ev-softmax in terms of time and space, relative to other sparse normalization alternatives?
- One of the motivations posed in the intro is multimodality, especially in the context of machine translation where multiple reference words are likely. However, in the experiments, this effect hasn't really been validated. The function has instead been applied to the attention distribution.


Other comments:
- In Sec 2.1, could you expand upon your earlier claim that the Gumbel-Softmax estimator introduces bias?
- Some citations / comparisons to other relevant work such as Zhao et al (Explicit Sparse Transformer: Concentrated Attention Through Explicit Selection) are missing.
- Line 226: appears to be a typo.

Overall, I think the paper is written well and has good motivation. The experiments have some weaknesses highlighted above and performance improvements seem small. Also, the description of the intuition behind the approach (why it induces multimodality + sparsity) would be useful.


**Time Spent Reviewing:**

4

---

> ### Author Response · Authors · 2021-08-10
> **Response and Clarifications to Reviewer fAmj (part 1)**
>
> We thank the reviewer for their thorough and thoughtful review. We address each question/concern individually:
>
> 1. > I'm not sure I completely understand the advantage of applying evidential softmax at training time.
>
> The advantage of applying evidential softmax at training time is to enable the the models to be trained directly with sparse latent distributions, with the expectation that this will lead to better performance. For VAE and CVAE models, for example, the decoders can be trained with sparse inputs, whereas with the post-hoc method in Itkina et al. [10] they would be trained with dense inputs. For the transformer model, this effect is even more significant, as each self-attention layer of the transformer is used as input to the next layer. Intuitively, training with a sparse distribution in this case enables the self-attention layers to provide nonzero attention on relevant words. We evaluated the semi-supervised VAE and translation experiments with evidential softmax applied only at test time, achieving an accuracy of $97.01 \pm 0.001$ in the semi-supervised VAE experiment and a BLEU score of 29.2 in the translation experiment (more metrics below). These results corroborate the claim that applying evidential softmax at training time improves the performance of models on a variety of tasks, compared to applying evidential softmax at only test time.
>
> 2. > Currently, it might not be giving enough intuition to the readers about why ev-softmax gives sparse \& multimodal distributions.
>
> Ev-softmax provides sparse and multimodal distributions by utilizing evidential theory to distinguish between lack of evidence and conflicting evidence. The intuition is that conflicting evidence is a source of multimodality, whereas a lack of evidence is a source of sparsity. Concretely, our function automatically identifies a threshold for the softmax function for which inputs below this threshold will be assigned zero probability. We will update the paper to include more of this explanation.
>
> 3. > Re: the MNIST experiments in sec 4.1, both sparsemax and entmax-1.5 seem to have degenerate performance in this experiment but they perform reasonably well in the other image generation experiments. Do you know why this is the case?
>
> Our experience with sparsemax and the entmax family is that they collapse multimodality in smaller latent spaces (like 10 in MNIST for example), but this is less pronounced in larger latent spaces as frequency of unimodal distributions decreases. Therefore, the performance in the VQ-VAE experiments is reasonably well for all the methods. In the semi-supervised VAE experiment, the number of decoder calls, which represents the number of latent classes assigned a nonzero probability, is also very close to 1 for both sparsemax and entmax-$1.5$. However, the small proportion of labeled data guides the latent classes, unlike the case of the MNIST experiments in Section 4.1, allowing for reasonable performance. One potential follow-up experiment (as suggested by Reviewer 1) is to examine the performance of these models with smaller proportions of labeled data and more complex datasets.
>
> 4. > Did you consider varying the value of alpha in entmax? If yes, does that help improve the multimodality of the distributions?
>
> We refer the reviewer to Figure 4 of Peters et al. [16], which shows the effect of varying the value of $\alpha$ on validation accuracy. These figures demonstrate that the validation accuracy is not significantly affected by the value of $\alpha$ for attention, and only slightly affected by the value of $\alpha$ for the output distribution of target words (between $\alpha = 1.5$ and the optimal value of $\alpha = 1.25$). We use $\alpha = 1.5$ due to its balance between performance and ease of computation.
>
> 5. > How well does the wide residual network used for evaluating the models in the exp in 4.3 perform? Instead of using a model for evaluation, did you consider performing human evaluation on a smaller sample?
>
> The wide residual network obtained a top-5 accuracy of 84% on the validation set. We chose to use a wide residual network over human evaluation to reduce the risk of bias in the experiment. We thank the reviewer for a good suggestion, and we leave human evaluation of images generated by models trained on larger datasets as future work.
>
> 6. > The improvements on the machine translation experiments appear to be very small. Have you performed significance testing (perhaps using a paired bootstrap test)? Computing multiple automatic metrics such as ROUGE, METEOR and performing human evaluation would be useful too.
>
> We thank the reviewer for the helpful suggestions here. The focus of this paper was on the novel method and demonstrating its applicability on a large scale, especially its ability to yield distributions that are more multimodal in nature than existing alternatives. Following the reviewer's suggestion, we computed ROUGE and METEOR scores, as well as a significance test, on the translations as follows:
>
> |                    | ROUGE-1 | ROUGE-2 | ROUGE-L | METEOR | p-value |
> | ------------------ | ------- | ------- | ------- | ------ | ------- |
> |Softmax	         | 59.31   | 35.62   | 56.09   | 57.02  | $< 0.05$ |
> |Post-hoc evidential | 59.09   | 35.42   | 55.93   | 56.84  | $< 0.01$ |
> |Sparsemax           | 58.47   | 34.76   | 55.39   | 56.33  | $< 0.001$ |
> |Entmax-1.5          | 58.94   | 35.20   | 55.75   | 56.83  | $< 0.01$ |
> |**Ev-softmax (ours)**  | **59.32** | **35.74** | **56.18** | **57.20** | N/A |
>
> Here, we compute the significance test as a one-sided paired bootstrap test between the METEOR scores of each normalization function and the ev-softmax function.
>
> Since our method balances sparsity and multimodality, it is able to perform closely (but still better) than softmax, and captures more important modes than the aggressively sparse distributions of sparsemax and entmax-1.5. We also see that the post-hoc evidential sparsification underperforms softmax, which is reasonable as the model was trained with softmax as the normalization functions in the self-attention layers.
>
> 7. > How can one guarantee that it is multimodality that is helping performance? I understand the \# attended words is higher / latent space reduction is (sometimes) higher with ev-softmax. However, there is lack of evidence showing that multimodality helps performance.... One of the motivations posed in the intro is multimodality, especially in the context of machine translation where multiple reference words are likely. However, in the experiments, this effect hasn't really been validated. The function has instead been applied to the attention distribution.
>
> While we cannot guarantee that multimodality is helping performance due to the nature of the task, we provide experiments that are multimodal in nature. The toy experiment of the MNIST image generation, conditional on an even or odd input, demonstrates one case in which the sparse yet multimodal distributions are necessary to perform well. In this experiment, sparsemax and entmax-1.5 yield degenerate prior distributions, and it is clear that the multimodality helps performance here.
>
> 8. > How expensive is the ev-softmax in terms of time and space, relative to other sparse normalization alternatives?
>
> We did not perform rigorous benchmarking of the computational cost of ev-softmax relative to the other sparse normalization alternatives as our experiments were run on machines shared with other researchers. However, our implementation of ev-softmax is almost identical to the PyTorch implementation of softmax, and we observed empirically that training the machine translation models with ev-softmax required roughly the same time as training those with softmax. On the WMT German-English dataset using a single NVIDIA GeForce 1080 GPU, Peters et al. [16] report a processing speed of 13,000 words per second for 1.5-entmax, compared with 14,500 words per second for softmax. The publicly available implementations of sparsemax and entmax-$1.5$ are $O(d \log d)$, where $d$ is the number of dimensions of the input, though an $O(d)$ algorithm exists for sparsemax [8]. In comparison, our implementation of ev-softmax has a time complexity of $O(d)$. All sparse normalization techniques require $O(d)$ space.
>
> 9. > One of the motivations posed in the intro is multimodality, especially in the context of machine translation where multiple reference words are likely. However, in the experiments, this effect hasn't really been validated. The function has instead been applied to the attention distribution.
>
> While it is difficult to validate this claim systematically on a large scale due to the nature of the task, we demonstrate in Fig. 3 one example in which a multimodal attention distribution enables a more accurate translation in a scenario which required the context of multiple words to produce the correct translation.

---

> > ### Author Response · Authors · 2021-08-10
> > **Response and Clarifications to Reviewer fAmj (part 2)**
> >
> > 10. > In Sec 2.1, could you expand upon your earlier claim that the Gumbel-Softmax estimator introduces bias?
> >
> > We refer the reviewer to Section 2 of Paulus, Maddison, and Krause, "Rao-Blackwellizing the Straight-Through Gumbel-Softmax Gradient Estimator." The Gumbel-Softmax estimator is a biased estimator of the true gradients because the Gumbel-Softmax relaxation results in a distribution that is not identical to the true categorical distribution. Concretely, given a continuously differentiable $f : \mathbb{R}^{2n} \to \mathbb{R}$, $\theta \in \mathbb{R}^n$ and a discrete one-hot random variable $D \in${$0, 1$}$^n$ with $\sum_i D_i = 1$ and distribution given by $p_{\theta}(D) \propto e^{D^T \theta}$, consider the minimization problem $\min_{\theta}\mathbb{E}[f(D, \theta)]$. Then for the tempered softmax function $h_{\tau}: \mathbb{R}^n \to \mathbb{R}^n$ given by $h_{\tau}(x) \propto e^{x / \tau}$, define the random variable $S_{\tau} = h_{\tau}(\theta + G)$, where $G$ is a vector of i.i.d. $G_i \sim \text{Gumbel}$ random variables. Then $D = \lim_{\tau \to 0} S_{\tau}$ and the gradient of the relaxed loss $\nabla_{\theta} \mathbb{E}[f(S_{\tau}, \theta)]$ can be reparameterized with the gradient estimator $\nabla_{GS} := \frac{\partial f(S_\tau, \theta)}{\partial S_{\tau}} \frac{dS_{\tau}}{d\theta}$. This gradient estimator is a biased estimator of $\nabla_{\theta}\mathbb{E}[f(D, \theta)]$ for $\tau > 0$.

---

> > > ### Comment · Reviewer_fAmj · 2021-08-20
> > > **Response to rebuttal**
> > >
> > > Dear authors,
> > >
> > > Thank you for your response. Below I'm following up on the points that I have further comments/questions about:
> > >
> > > 1. I understand that applying evidential softmax at training time is done with the goal of performance improvements. However, I think the paper is lacking discussion about why this outcome is expected. Can you characterize the differences in terms of the output attention distributions? Are they sparser / more multimodal compared to the post-hoc approach of Itkina et al?
> > >
> > > 3./4. I think it might be valuable in understanding previous approaches like sparsemax and entmax a bit more. The authors mentioned that these normalization functions “collapse multimodality in smaller latent spaces". Maybe I missed something but I'm not sure why this would naturally be the case, and the paper doesn't dig deeper in understanding this. Similarly, I think the authors could’ve done a sweep over different values of alpha for entmax. The plot from Peters et al that the authors refer to is based on their MT experiment, but the trend could very well be different for other tasks. I also think the follow-up (from R1) the authors mention could have been really valuable in understanding the pros/cons of these methods in different settings.
> > >
> > > 6. It seems that the improvements relative to Itkina et al and entmax are quite small on both image generation and machine translation. I really think significance testing is necessary across all settings, not only for the METEOR/ROUGE scores the authors reported above.
> > >
> > > Overall, I think the paper could be improved in understanding the shortcomings of previous work better, more clearly describing the benefits of ev-softmax at training time (both empirically and while motivating their approach) and significance testing their improvements.
> > > For these reasons, I think I will keep my current score.

---

> > > > ### Author Response · Authors · 2021-08-26
> > > > **Additional Response**
> > > >
> > > > We thank the reviewer for the response, and we address the questions/comments here:
> > > > > 1. I understand that applying evidential softmax at training time is done with the goal of performance improvements. However, I think the paper is lacking discussion about why this outcome is expected. Can you characterize the differences in terms of the output attention distributions? Are they sparser / more multimodal compared to the post-hoc approach of Itkina et al?
> > > >
> > > > The motivation for applying evidential softmax at training time is that decoders and additional layers downstream of the sparse activation function can be trained with sparse inputs. This benefits the performance in two ways. First, the weights of the downstream layers can be optimized for sparse inputs. Second, the losses are back-propagated through the evidential softmax to allow for the weights upstream of the sparse layers to be optimized. From an optimization perspective, applying evidential softmax at test time may introduce bias, as the model was not optimized with the evidential softmax function, and training the model with evidential softmax resolves this problem.
> > > >
> > > > As such, comparing the output attention distributions when applying evidential softmax at test time versus those when applying evidential softmax at training time could be useful for building intuition, but it is difficult to make conclusive statements from such analysis since the two are different in nature.
> > > >
> > > > > 3./4. I think it might be valuable in understanding previous approaches like sparsemax and entmax a bit more. The authors mentioned that these normalization functions “collapse multimodality in smaller latent spaces”. Maybe I missed something but I’m not sure why this would naturally be the case, and the paper doesn’t dig deeper in understanding this. Similarly, I think the authors could’ve done a sweep over different values of alpha for entmax. The plot from Peters et al that the authors refer to is based on their MT experiment, but the trend could very well be different for other tasks. I also think the follow-up (from R1) the authors mention could have been really valuable in understanding the pros/cons of these methods in different settings.
> > > >
> > > > Comparing sparsemax and evidential softmax, we claim the following:
> > > >
> > > > **Theorem:** For any positive integer $n$ and $\mathbf{x} \in \mathbb{R}^n$, $\texttt{evsoftmax}(\mathbf{x})_j = 0$ implies that $\texttt{sparsemax}(\mathbf{x})_j = 0$.
> > > >
> > > > **Proof:** Consider arbitrary $n$-dimensional vector $\mathbf{x} \in \mathbb{R}^n$. Without loss of generality, since sparsemax and evidential softmax are invariant to translations, we can transform $\mathbf{x} \mapsto \mathbf{x} - \frac{1}{n}\sum_{i=1}^n x_i + \frac{1}{n}$ so that $\sum_{i=1}^n x_i = 1$. Then since $\texttt{sparsemax}$ is defined as the $\ell_2$ projection onto the simplex, we can see by inspection that $\texttt{sparsemax}(\mathbf{x})_j= 0$ iff $x_j \le 0$.
> > > >
> > > > Now for the same input, evidential softmax is defined such that $\texttt{evsoftmax}(\mathbf{x})_j = 0$ iff $x_j$ is less than the mean of the elements of $\mathbf{x}$, which is $\frac{1}{n}$. That is,  $\texttt{evsoftmax}(\mathbf{x})_j = 0$ iff $x_j < \frac{1}{n}$.
> > > >
> > > > Comparing these two statements, we see that evidential softmax and sparsemax both apply a threshold for rendering outputs sparse, but sparsemax's threshold is lower by $\frac{1}{n}$. Note that this holds true for any arbitrary $n$ and $\mathbf{x} \in \mathbb{R}^n$, which will simply have these thresholds adjusted by the same constant of $\frac{1}{n} \sum_{i=1}^n x_i - \frac{1}{n}$. Thus, it follows that $\texttt{evsoftmax}(\mathbf{x})_j = 0$ implies $\texttt{sparsemax}(\mathbf{x})_j = 0$. $\blacksquare$
> > > >
> > > > This matches the empirical observation that sparsemax generates distributions that are sparser than those of evidential softmax. Furthermore, note that the difference in the thresholds is $\frac{1}{n}$, which decreases as $n$ increases. Thus, as the number of dimensions increases, the difference in thresholds for sparsity given by evidential softmax and sparsemax is decreased. This explains why the difference in sparsity between sparsemax and evidential softmax is less pronounced as the number of dimensions increases. It is difficult to generalize this behavior across arbitrary values of $\alpha$ for the entmax function.
> > > >
> > > > We will include this analysis in the updated paper.
> > > >
> > > > Edit: The response below provides a counterexample for the claim above, which is mistaken. We thank the area chair for their correction.
> > > >
> > > > > 6. It seems that the improvements relative to Itkina et al and entmax are quite small on both image generation and machine translation. I really think significance testing is necessary across all settings, not only for the METEOR/ROUGE scores the authors reported above.
> > > >
> > > > We thank the reviewers for this suggestion. We include standard errors on all but the VQ-VAE experiments, which we originally omitted due to computational limits but are working on computing.

---

> > > > > ### Comment · Reviewer_fAmj · 2021-08-29
> > > > > **Follow up**
> > > > >
> > > > > Thanks for your response. Re. the softmax/ev-softmax analysis, I think this could be helpful to include in the paper.
> > > > >
> > > > > I still don't have a great sense of why a direct comparison with Itkina et al is not provided in this work. This could have verified that applying evidential softmax at training time is valuable. Similarly, another comparison that could've been worth including would have been sweeping over different values of alpha for entmax, even just for one of the experiments. I think the paper could have benefitted by having these baselines.

---

> > > > > > ### Author Response · Authors · 2021-08-30
> > > > > > **Follow Up Response**
> > > > > >
> > > > > > Thanks for the suggestion to add direct comparison with Itkina et al. We performed these comparisons as part of the rebuttals (10. in rebuttal to R1 and 6. in rebuttal to R2) and will include the results in the updated paper.

---

> > > > > ### Comment · Area_Chair_bETE · 2021-09-05
> > > > > **Incorrect claim?**
> > > > >
> > > > > Dear authors,
> > > > >
> > > > > I don't think the "theorem" in your previous answer is correct. You say
> > > > >
> > > > > "we can see by inspection that sparsemax(x)_j  = 0 iff x_j <= 0" (assuming sum(x) = 1)
> > > > >
> > > > > I don't get how we can see this "by inspection", and in fact here is a counterexample:
> > > > >
> > > > > x = [2, 0.5, -1.5] => sparsemax(x) = [1, 0, 0]
> > > > >
> > > > > Your claim that ev_softmax(x)_j  = 0 => sparsemax(x)_j = 0 does not seem to be correct.
> > > > >
> > > > > Here's a counterexample:
> > > > >
> > > > > x = [0.8, 0.2, 0] => sparsemax(x) = [0.8, 0.2, 0]
> > > > >
> > > > > For this x, since 0.2 <= 1/3 ev_softmax(x)_2 = 0 but sparsemax(x)_2 = 0.2 > 0.
> > > > >
> > > > > Please let me know if there is something I'm missing.
> > > > >
> > > > > Thanks.

---

> > > > > > ### Author Response · Authors · 2021-09-05
> > > > > > **Correction**
> > > > > >
> > > > > > Thank you for the response. You are correct that the statement "we can see by inspection that sparsemax(x)_j = 0 iff x_j <= 0" (assuming sum(x) = 1) is inaccurate. The corrected statement is that $x_j \le 0$ implies $\texttt{sparsemax}(x)_j = 0$. Assuming $\sum_i x = 1$, then $x$ lies outside the simplex in the hyperplane containing the simplex $\sum_i x = 1$, and in this hyperplane, the convexity of the simplex and the convexity of the Euclidean distance imply that the optimal solution will lie on the boundary of the simplex. Among the boundary of the simplex, then one can see that the optimal solution should lie in the boundary $\texttt{sparsemax}(x)_j = 0$.

---

> > > > > > > ### Comment · Area_Chair_bETE · 2021-09-06
> > > > > > > **Thanks for clarifying.**
> > > > > > >
> > > > > > > Thanks for the quick clarification. Your corrected statement seems accurate. For the record, note however that the "theorem" ev_softmax(x)_j = 0 => sparsemax(x)_j = 0 is not correct, and in fact the two counterexamples above show that the supports of ev_softmax and sparsemax are generally not a subset of each other:
> > > > > > >
> > > > > > > x = [2, 0.5, -1.5] => sparsemax(x) = [1, 0, 0], ev_softmax(x) = [>0, >0, 0]
> > > > > > >
> > > > > > > x = [0.8, 0.2, 0] => sparsemax(x) = [0.8, 0.2, 0], ev_softmax(x) = [>0, 0, 0]
> > > > > > >
> > > > > > > I don't think this compromises the main findings of the paper, but it should impact the discussion about the relation between the two transformations suggested by one of the reviewers.

---

### Official Review · Reviewer_kvJZ · 2021-07-16

**Rating:** 4
**Confidence:** 3

**Summary:**

The paper proposes a new sum-to-one activation function, namely evidential softmax, which is specialized for the sparse multimodal activation.
With the suggested transformation, the authors resolve the existing post-hoc process of evidential sparsification.
The authors provide various experiments to support their proposed sparse activation function.



**Limitations And Societal Impact:**

The authors pointed out the limitation of their work, and the work seems that it does not have any potential negative societal impact.


**Main Review:**


Originality:
Removing the post-hoc process and finding the equivalent form seems quite novel work.
However, I'm a bit confused of the messeage of the paper.
The motivation in the introduction and the final purpose that can be seen in the experimental result section of the paper seems quite distinguished.

Quality:
The paper sound okay technically, however, I'm not convinced with the experimental section and the related questions are separately listed below.

Clarity:
I cannot say that the paper is well-written due to the overall configuration.
I wonder why the discrete VAE came up in the first in the background.
It would be better if Section 5 come before Section 4.

Significance:
The paper is explained through theoretical backups.
However, the overall performance is too marginal and Table 2 seems to be taken from [9] but there are no comment on this.
Also, there is no confidence interval in VQ-VAE experiment.
As a minor comment, it would be more convinced if the authors applied various \alpha values for the entmax.

Questions:
- In Table 1, compared to the other sum-to-one activation functions, EvSoftmax has less nice properties. How does the authors think that which property unsatisfaction gives better result?
- How does the result change in Fig 1 if the ratio over the labels is unbalanced? (including the extremely unbalanced case)
- What does decoder calls imply? And is it good or bad if the number is small?
- SS-VAE experiment result shows little difference across various models. What becomes when using extremely small amount of labeld data? (for example, 1%) Also, how does the performance gap change when using more complex datasets such as OMNIGLOT or CIFAR?
- How is the top-1 accuracy result in the VQ-VAE experiment?
- How does overall performance change as \epsilon differs?
- I can't find any comparison on the post-hoc process [10] in the experiment section. How differ the performance are?


**Time Spent Reviewing:**

8

---

> ### Author Response · Authors · 2021-08-10
> **Response and Clarifications to Reviewer kvJZ**
>
> We thank the reviewer for their careful reading and thoughtful comments. We address each point individually:
>
> 1.  > The motivation in the introduction and the final purpose that can be seen in the experimental result section of the paper seems quite distinguished.
>
> While the nature of the large-scale tasks prevents a thorough investigation of how multimodality benefits performance, our experiments provide a wide range of tasks to show that our method benefits performance in tasks that are sparse and multimodal in nature, such as conditional image generation and language modeling. Furthermore, the MNIST conditional image generation experiment provides a concrete example of how the combination of multimodality and sparsity can yield a latent variable model with latent classes which reflect our intuition for the "true" latent classes of the MNIST digits. In terms of the tractability of marginalization over large latent spaces, we present this paper as an investigation of the evidential softmax function on a wide range of tasks, and we leave the application to larger tasks as future work. We do not perform rigorous benchmarking of the computational cost of ev-softmax relative to other alternatives as our experiments were run on machines shared with other researchers. However, we report the number of nonzero latent classes in the conditional image generation experiments as a proxy for the expected improvement in tractability.
>
> 2. > I wonder why the discrete VAE came up in the first in the background. It would be better if Section 5 come before Section 4.
>
> We identified discrete VAEs, normalization functions, and the post-hoc evidential sparsification function introduced in Itkina et al. [10] as three topics to discuss in background before presenting the evidential softmax transformation. We felt that normalization functions naturally led to an introduction of the post-hoc function, which naturally led to our generalization in Section 3. The remaining position in background for discrete VAEs was therefore in the beginning.
>
> In our background, we had covered the related work which we had felt were most relevant to our experiments, namely alternative sparse normalization functions [8, 10, 16-18] and stochastic estimators of gradients through discrete spaces [14-15, 20-21]. Therefore, we felt that the related work could be positioned following the experiments, given that the background had covered the most relevant related work.
>
> 3. > There is no confidence interval in VQ-VAE experiment.
>
> The VQ-VAE experiment was performed with only 1 random seed due to the computational cost of the experiment. We will add confidence intervals to the updated paper.
>
> 4. > It would be more convinced if the authors applied various $\alpha$ values for the entmax.
>
> We refer the reviewer to Figure 4 of Peters et al. [16], which shows the effect of varying the value of $\alpha$ on validation accuracy on a translation task. These figures demonstrate that the validation accuracy is not significantly affected by the value of $\alpha$ for attention, and only slightly affected by the value of $\alpha$ for the output (between $\alpha = 1.5$ and the optimal value of $\alpha = 1.25$).
>
> 5. > Table 2 seems to be taken from [9] but there are no comment on this.
>
> In lines 259-260, we report that the results in Table 2 for the stochastic methods are obtained from Correia et al. [9].
>
> 6. > How does the authors think that which property unsatisfaction gives better result?
>
> The properties that evidential softmax does not satisfy include idempotence and scale invariance. While we do not claim that either property dissatisfaction is an advantage, we feel that neither property is very applicable in the setting of normalization functions that map real vectors to probability distributions. As detailed in Section 3.1, normalization functions are generally only applied once, so idempotence is not relevant in this context. In addition, the scale of the inputs to normalization functions is often a learned feature of the data which provides a measure of confidence in predictions, and thus, scale invariance is not necessarily beneficial in this context.
>
> 7. > What does decoder calls imply? And is it good or bad if the number is small?
>
> The number of decoder calls is equivalent to the number of elements in the latent space assigned a nonzero probability. The significant implication of this is how costly it is to evaluate the VAE, which is a measure of how much of a latent space reduction is achieved.
>
> 8. > How is the top-1 accuracy result in the VQ-VAE experiment?
>
> As a relatively small dataset, tinyImagenet contains 500 images of size $64 \times 64$ for each of the 200 classes. As such, the top-1 accuracy on generated images is a challenging task given the relatively small number of training examples and image sizes, as well as the relatively large number of classes. The top-5 accuracy is proposed as a metric in "Classification Accuracy Score for Conditional Generative Models" by Ravuri and Vinyals, and it is also reported in Razavi, Oord, and Vinyals [37] and Itkina et al. [10]. Furthermore, we use the top-5 and top-10 accuracies given the intuition that maintaining higher multimodality in the latent classes results in a higher likelihood that the features of the correct class are captured in the latent space. Furthermore, high classification accuracy is not our objective; rather, it is a means to compare the different normalization functions. Thus, the relative numbers are more relevant than the absolute values here.
>
> 9. > How does overall performance change as $\epsilon$ differs?
>
> Empirically, varying $\epsilon$ between $\epsilon \in \{10^{-7}, 10^{-6}, 10^{-5}\}$ did not change the results of our experiments.
>
> 10. > I can't find any comparison on the post-hoc process [10] in the experiment section. How differ the performance are?
>
> We do not report comparison to the post-hoc process presented in Itkina et al. [10] as we compare only to other methods which can be applied at both train and test time. Our proposed method is functionally equivalent to the post-hoc sparsification process, and therefore enabling training is a strict improvement over only performing the post-hoc process. Following the reviewer's comment, we evaluated the semi-supervised VAE and the translation models with the post-hoc process. For the semi-supervised VAE, the post-hoc process achieved an accuracy of $97.01 \pm 0.001$, and for the translation task, the post-hoc process achieved a BLEU score of 29.2 (more metrics are provided in the response for reviewer 2 below). In both of these cases, the post-hoc process performs around the same as softmax, which matches the claims in Itkina et al. [10], whereas our evidential softmax function outperforms both the post-hoc process and softmax.

---

### Decision · Program_Chairs · 2021-09-27

**Decision:**

Accept (Poster)

**Comment:**

The paper introduces a normalizing function called evidential softmax (ev-softmax). Based on principles of evidential theory, ev-softmax is able to assigns zero probability to some classes. It extends Itkina et al. (2020) by recovering their transformation and allowing its usage at training time. The experiment section shows (relatively small) performance gains compared to other sparse methods on several simple tasks and on a machine translation task.

Most reviewers agree that this is a good paper that proposes an interesting contribution, closes the loop with Itkina et al. (2020),  and establishes an interesting link between evidential theory and sparse transformations. The main weaknesses which have been pointed out are  lack of details about some of the experiments (clarified in the rebuttal), lack of camparison with alpha-entmax for other values of alpha (not crucial but a nice-to-have), and lack of discussion of how ev_softmax relates to other sparse transformations (which the authors elaborate on in the rebuttal, but could be further expanded). This changes seem doable at camera ready time, therefore I recommend acceptance.

I urge the authors to follow the recommendations of the reviewers. Reporting other values of alpha should be simple given that the code for alpha-entmax is publicly available. More importantly, the relation between ev_softmax and entmax/sparsemax should be clarified:
- The claim that ev_softmax(x)_j = 0 => sparsemax(x)_j = 0 is not correct - the supports of ev_softmax and sparsemax are generally not a subset of each other (check comment with counterexamples)
- Note that the support of sparsemax can also be expressed as a mean condition on the largest scores — see Alg 1 in https://arxiv.org/pdf/1602.02068.pdf. The zeros of sparsemax are the entries where x_j <= mean(top_k(x)) - 1/k, where k is the size of the support. I think this deserves discussion.
- When x in R^2, sparsemax is a hard sigmoid (Fig 1 in the ref above) but ev_softmax seems to be a 0/1 loss, piecewise constant, with zero gradients everywhere. This means you can’t backpropagate through ev_softmax for this 2-dimensional case. I think this an important limitation that should be discussed (sparsemax and entmax don’t have this limitation).